# Recent Advances in Electrochemical Sensors for Detecting Toxic Gases: NO_2_, SO_2_ and H_2_S

**DOI:** 10.3390/s19040905

**Published:** 2019-02-21

**Authors:** Md Ashfaque Hossain Khan, Mulpuri V. Rao, Qiliang Li

**Affiliations:** Department of Electrical and Computer Engineering, George Mason University, Fairfax, VA 22030, USA; rmulpuri@gmu.edu

**Keywords:** gas sensor, nitrogen dioxide (NO_2_), sulphur dioxide (SO_2_), hydrogen sulfide (H_2_S), density-functional theory (DFT), Internet of Things (IoT), sensitivity, response/recovery time

## Abstract

Toxic gases, such as NO_x_, SO_x_, H_2_S and other S-containing gases, cause numerous harmful effects on human health even at very low gas concentrations. Reliable detection of various gases in low concentration is mandatory in the fields such as industrial plants, environmental monitoring, air quality assurance, automotive technologies and so on. In this paper, the recent advances in electrochemical sensors for toxic gas detections were reviewed and summarized with a focus on NO_2_, SO_2_ and H_2_S gas sensors. The recent progress of the detection of each of these toxic gases was categorized by the highly explored sensing materials over the past few decades. The important sensing performance parameters like sensitivity/response, response and recovery times at certain gas concentration and operating temperature for different sensor materials and structures have been summarized and tabulated to provide a thorough performance comparison. A novel metric, sensitivity per ppm/response time ratio has been calculated for each sensor in order to compare the overall sensing performance on the same reference. It is found that hybrid materials-based sensors exhibit the highest average ratio for NO_2_ gas sensing, whereas GaN and metal-oxide based sensors possess the highest ratio for SO_2_ and H_2_S gas sensing, respectively. Recently, significant research efforts have been made exploring new sensor materials, such as graphene and its derivatives, transition metal dichalcogenides (TMDs), GaN, metal-metal oxide nanostructures, solid electrolytes and organic materials to detect the above-mentioned toxic gases. In addition, the contemporary progress in SO_2_ gas sensors based on zeolite and paper and H_2_S gas sensors based on colorimetric and metal-organic framework (MOF) structures have also been reviewed. Finally, this work reviewed the recent first principle studies on the interaction between gas molecules and novel promising materials like arsenene, borophene, blue phosphorene, GeSe monolayer and germanene. The goal is to understand the surface interaction mechanism.


**Contents**



IntroductionRecent Advances in NO_2_ Gas Detection2.1.Graphene and Its Derivatives-Based NO_2_ Sensors2.2.Transition Metal Dichalcogenide (TMD)-Based NO_2_ Sensors2.3.Metal and Metal-Oxide Nanostructure-Based NO_2_ Sensors2.4.GaN-Based NO_2_ Sensors2.5.Organic Materials-Based NO_2_ Sensors2.6.Hybrid Materials-Based NO_2_ SensorsRecent Advances in SO_2_ Gas Detection3.1.Carbon Material-Based SO_2_ Sensors3.2.Metal and Metal-Oxide Nanostructures-Based SO_2_ Sensors3.3.GaN-Based SO_2_ Sensors3.4.Solid Electrolyte-Based SO_2_ Sensors3.5.Zeolite-Based SO_2_ Sensors3.6.Paper-Based SO_2_ SensorsRecent Advances in H_2_S Gas Detection4.1.Carbon Material-Based H_2_S Sensors4.2.GaN-Based H_2_S Sensors4.3.Metal and Metal Oxide-Based H_2_S Sensors4.3.1.Nanostructured Metal Oxide-Based Sensors4.3.2.Mesoporous Metal Oxide-Based Sensors4.3.3.Metal Oxide Microsphere-Based Sensors4.4.MOF-Based H_2_S Sensors4.5.Organic Materials-Based H_2_S Sensors4.6.Solid Electrolytes-Based H_2_S SensorsRecent Density-Functional Theory (DFT) Study of Gas Molecule-Sensor InteractionCalibration of Toxic Gas SensorsToxic Gas Sensors in Internet of Things (IoT) ApplicationsFuture Perspectives and Conclusions


Acknowledgments

Conflicts of Interest

References

## 1. Introduction

Humans are exposed to various air toxins in the indoor and outdoor environment. Poor air quality is a well-known trigger for various health problems which can often result in life threatening and expensive emergency care. Therefore, precise toxic gas sensing will not only bring a major benefit to industries but also to day-to-day life for all people. Nitrogen dioxide (NO_2_) is one of the common toxic air pollutants, which is mostly found as a mixture of nitrogen oxides (NO_x_) with different ratios (x). NO_2_ is a reddish-brown, irritant, toxic gas having a characteristic sharp and biting odor. The LC_50_ (the lethal concentration for 50% of those exposed) for one hour of NO_2_ exposure for humans has been estimated as 174 ppm. The major sources of NO_2_ are from combustion of fuels such as certain coals and oil [1], biomass burning due to the extreme heat of lightning during thunderstorms [2], and nitrogen fixation by microorganisms due to agricultural fertilization [3]. The noteworthy impacts of NO_2_ include: respiratory inflammation of the airways, decreased lung function due to long term exposure, increased risk of respiratory conditions [4,5], increased responsiveness to allergens, contribution to the formation of fine particulate matter (PM) and ground level ozone which have adverse health effects, and contribution to acid rain causing damage to vegetation, buildings and acidification of lakes and streams [6,7]. 

Sulphur dioxide (SO_2_) is the most common air pollutant, mostly found as a mixture of sulfur oxides (SO_x_). It is an invisible gas with a nasty, sharp smell. The maximum concentration for SO_2_ exposures of 30 min to 1 h has been estimated as 50 to 100 ppm. The main sources of SO_2_ include burning of fossil fuels (fuel oil, coal) in power stations, oil refineries, other large industrial plants, motor vehicles and domestic boilers [8,9]. It is also produced from natural sources like active volcanoes and forest fires. When mineral ores containing sulfur are processed, SO_2_ is released to atmosphere as well. Excessive exposure of SO_2_ causes harms on the eye, lung and throat [10,11]. It is toxic to some plants, inducing visible signs of injury and reducing yields. SO_2_ gas combined with air moisture causes gradual damage to some building materials (e.g., limestone). SO_2_ can readily dissolve in the water droplets in clouds, causing acid rain that affects natural balance of rivers, lakes and soils, resulting in damage to wildlife and vegetation. 

Hydrogen sulfide (H_2_S) is a highly toxic, malodorous, intensely irritating gas. The maximum concentration for H_2_S exposure for one hour without grave after-effects has been estimated as 170 to 300 ppm. The key sources of H_2_S gas are from decaying organic materials, natural gas, volcanic gas, petroleum, sewage plants and sulfur deposits [12,13]. Minimal exposure to H_2_S gas causes nose/eye irritation, olfactory nerve paralysis. Moderate amount may cause sore throat, cough, keratoconjunctivitis, chest tightness and pulmonary edema. Excessive exposure causes headaches, disorientation, loss of reasoning, coma, convulsions and even death [14,15]. 

In comparison to gas detection techniques like optical [16], acoustic [17] and gas chromatographic methods [18], electrochemical sensing is the most popular technique for ambient toxic gas monitoring. The key advantages of electrochemical detection are having low energy linear output with high resolution, good selectivity and repeatability, ppm level detection with high accuracy, and being more inexpensive than other techniques [19]. However, electrochemical sensors are highly sensitive to temperature fluctuations and have minimal shell life. The operating temperature should be kept as steady as possible to get the best sensor performances. Sensors with high operating temperature are generally employed in industrial and space applications. Over the last decades, research on toxic gas sensing was mostly focused on using electrochemical sensors which were built from various functional materials, such as carbon nanomaterials [20,21,22,23,24,25,26,27,28,29,30,31], metal oxide/metallic nanostructures [32,33,34,35,36,37,38], transition metal dichalcogenides (TMDs) [39,40,41], gallium nitride (GaN) [42,43,44,45], organic materials [46,47,48,49,50], solid electrolytes [51,52,53,54], zeolites [55,56,57,58] and others [59,60,61,62,63]. Recently numerous research efforts have been made on suitable gas sensing materials to detect nitrogen oxides, sulfur oxides and hydrogen sulfide gases. In this work, we have reviewed the recent advances in electrochemical sensors for toxic gas detection focusing mainly on NO_2_, SO_2_ and H_2_S gas sensors. The recent progress of each of these toxic gas detections has been categorized based on various sensing materials of high interest. The goal is to shine a light on the future development trend of toxic gas detection, a vital technology for the emerging Internet of Things era. 

## 2. Recent Advances in NO_2_ Gas Detection

### 2.1. Graphene and Its Derivatives-Based NO_2_ Sensors

Graphene provides a large surface area, atom-thick 2D conjugated structures, low electrical noise, high conductivity, and excellent electronic properties [64]. Having high surface-to volume ratio, reduced graphene oxide (RGO) provides large surface areas, defects and low electrical noise as well [65]. All these characteristics make both graphene and RGO suitable candidates for gas adsorption and detection. 

Recently epitaxial graphene has been utilized to detect ppb levels of NO_2_ gas and it was found that single-layer graphene is superior to bilayer graphene in terms of carrier concentration response [66]. Wang et al. [67] incorporated Pd nanoparticles (NPs) and SnO_2_ NPs on reduced graphene oxide to form Pd-SnO_2_-RGO hybrids as NO_2_ gas sensing materials. A high resolution transmission electron microscopy (HR-TEM) image of Pd-SnO_2_-RGO reveals that 3–5 nm sized nanoparticles (NPs) are deposited on RGO nanosheets (Figure 1A). Lattice distances of 0.33 nm and 0.23 nm indicated the presence of SnO_2_ NPs and Pd NPs, respectively. When a Pd-SnO_2_-RGO nanosheet was exposed to 1 ppm of NO_2_ gas at room temperature, a response of 3.92 was obtained with a response time of 13 s which are better compared to RGO as well as SnO_2_-RGO hybrids. However, the recovery time (105 s) was slower due to addition of Pd NPs. To perform a concentration response test, the fabricated sensor was exposed to 50 ppb to 2 ppm of NO_2_ gas. The sensor showed an increasing response trend with NO_2_ concentration (Figure 1B). The selectivity response of the Pd-SnO_2_-RGO sensor was examined towards Cl_2_, NO and some common volatile organic compounds (VOCs). Results indicated that the Pd-SnO_2_-RGO hybrid is highly selective to NO_2_ gas (Figure 1C). Preferred adsorption sites providing for NO_2_, high conductivity and the catalytic properties of Pd NPs are mainly responsible for the sensing performance improvement. However, no sulfur-containing gas was included in the test interference gases. Since Pd is known to interact strongly with S [68], the fabricated sensor should have been tested with S-containing gases to get the complete selectivity test picture. The same Wang et al. group [69] experimented with the introduction of oxygen vacancies (OV) into reduced graphene oxide nanosheets decorated with SnO_2_ nanoparticles (NPs). OVs enhance the adsorption of O_2_ molecules which in turn enhances the adsorption of NO_2_ molecules onto SnO_2_ NPs. Upon exposure to 1 ppm of NO_2_ gas, the SnO_2_-RGO-OVs-based sensor showed a response of 3.80 with reasonable response and recovery time. These NO_2_ sensing performances are better than those of other previously reported RGO-based sensors. 

In another study, Akbari et al. [70] decomposed methane in an arc discharge experiment to get carbonaceous materials (C-strands) between graphite electrodes. Upon NO_2_ exposure, the conductivity of the fabricated C strands was altered due to charge transfer between the carbon film and NO_2_ molecules. Previously, Zhang et al. [71] reported a rGO/Au nanocomposite-based NO_2_ sensor using a hydrothermal treatment. It provided good sensitivity with a quick response–recovery process at 50 °C.

### 2.2. Transition Metal Dichalcogenide (TMD)-Based NO_2_ Sensors

Two-dimensional (2D) transition metal dichalcogenides (TMDs) possess semiconducting nature, high surface-to-volume ratio and atomically thin-layered structures which are useful properties required to be a convincing sensing material [72]. MoS_2_, WS_2_, ReS_2_, MoSe_2_, MoTe_2_, WSe_2_ and ReSe_2_ are very promising 2D TMDs for gas sensing purposes [73,74,75]. Agrawal et al. prepared in-plane and edge-enriched p-MoS_2_ flakes (mixed MoS_2_) to detect NO_2_ gas at room temperature [76]. A FE-SEM image of the mixed MoS_2_ flakes is shown in Figure 2A. The blackish region represents the in-plane MoS_2_ flakes and the white region represents the edge-enriched MoS_2_ flakes. Most likely, the edge-enriched MoS_2_ flakes are white due to their height from the substrate surface. Figure 2B displays a sensitivity vs. NO_2_ concentration bar graph at RT and 125 °C. NO_2_ is an electron acceptor and it withdraws electrons from the MoS_2_ flakes, thus causing the resistance decrease of the mixed MoS_2_ flake-based sensor. The response and recovery time of the sensor were better at 125 °C than at RT. This happened because the adsorption energy of the NO_2_ gas molecule with the MoS_2_ flakes is very high at RT. The sensitivity of the sensor had been enhanced under UV light illumination as shown in Figure 2C. This improvement is attributed to the photoactivated desorption of adsorbed oxygen and creation of fresh active sites on the edges of MoS_2_ flakes. In another study, Kumar et al. [77] prepared a 1D MoS_2_ nanowire network which showed a detection limit of 4.6 ppb NO_2_ with good sensitivity. At the estimated optimum operating temperature (60 °C), response and recovery times were found as 16 s and 172 s, respectively, at 5 ppm NO_2_ exposure. Previously, Choi et al. [78] introduced Nb atoms into 2D MoSe_2_ host films. Figure 2D displays the low magnification planar annular dark-field scanning transmission electron microscopy (ADF-STEM) images and FFT patterns (inset) of MoSe_2_: Nb 1C, where 1C indicates one deposition cycle in the plasma-enhanced atomic layer deposition (PEALD) process. The polycrystal ring patterns in the image represent the presence of a few grains. Variably Nb-doped MoSe_2_ sensor films were exposed to different NO_2_ concentrations as shown in Figure 2E. The highest gas response was found for a MoSe_2_:Nb 1C device among the three tested devices because at low Nb dopant concentrations, MoSe_2_ showed an improved NO_2_ gas response due to its small grains and stabilized grain boundaries. At high Nb dopant concentrations, the NO_2_ gas response was degraded due to the increase of gas-unresponsive metallic NbSe_2_ regions, so an optimum Nb concentration is required for achieving a better gas response. The resistance of the MoSe_2_-based sensor gradually increased due to oxidation, whereas the Nb-doped MoSe_2_ sensor showed very stable response (Figure 2F). This means, introduction of Nb atoms onto 2D layered MoSe_2_ promotes a stable gas response and the long-term stability of the sensor. Also, a significant enhancement in sensing response with quick response-recovery toward NO_2_ was observed on WS_2_ nanosheet functionalized with Ag NWs [79]. 

### 2.3. Metal and Metal-Oxide Nanostructure-Based NO_2_ Sensors

Metal oxides can be synthesized in various nanostructure morphologies like nanowires, nanoparticles, nanotubes, nanoflowers, nanocomposites and nanosheets for the enhancement of sensing performance [80,81,82]. Besides, porosity and permeable shell layers contribute to absolute electron depletion and gas diffusion that allow sensor devices to achieve high sensitivity toward gases [83]. Qiang et al. reported a NO_2_ gas sensor based on porous silicon (PS)/WO_3_ nanorods (NRs) functionalized with Pd NPs [84]. PS WO_3_ NRs were synthesized by electrochemical methods and thermal oxidation of W film, respectively. Pd NPs were deposited onto WO_3_ NRs, by the reduction of a Pd complex solution. Three different samples of PS/WO_3_ NRs–Pd NPs were prepared by varying the amount of Pd NPs on the substrate. These are PS/WO_3_–Pd20, PS/WO_3_–Pd40 and PS/WO_3_–Pd60, where the order of the amount of Pd NPs is Pd60 > Pd40 > Pd20. A TEM image of PS/WO_3_–Pd60 displays the agglomeration of Pd NPs on WO_3_ NRs (Figure 3A). Gas concentration tests on the PS/WO_3_–Pd60 sensor revealed a ppb level detection capacity at RT with a faster response time (Figure 3B). The catalytic activity of Pd NPs enhanced the NO_2_ molecule adsorption and thereby enhanced the sensor response, so a PS/WO_3_–Pd60 sensor having the highest amount of Pd NPs showed the highest sensor response at room temperature (Figure 3C). With a facile fabrication process and being compatible with the planar processes of the microelectronics industry, ultra-thin PdO films provided good sensing performances toward NO_2_ [85], but they require a long recovery period (600–700 s) because of the lack of immediate interaction between NO_2_ molecules and oxygen molecules adsorbed on sensor material surface. Also, ZnO nanostructured films obtained by a thermal evaporation method offered significantly enhanced response (622 at 100 ppm NO_2_) with good response-recovery at 200 °C [86]. The microwave-synthesized NiO film has been found to operate using ultra-low power of 0.2 μW at room temperature. It achieved a response of 4991% to 3 ppm NO_2_ along with fast response-recovery [87]. Moreover, a reasonable sensor response toward low concentration of NO_2_ was exhibited by the multicomponent oxide CuBi_2_O_4_ at 400°C [88]. Recently, Hung et al. synthesized three sensors of ZnO (Z2, Z4 and Z6) and Zn_2_SnO_4_ (ZS2, ZS4 and ZS6) NWs on microelectrode chips at 2, 4 and 6 cm from the thermal evaporation source, respectively [89]. It was found that the distance between the source and substrate strongly affected the gas response of the Zn_2_SnO_4_ NW sensors. Figure 3D,E show FESEM images of the on-chip grown ZnO (Z2) NW and Zn_2_SnO_4_ (ZS2) NW respectively. 

The sensing performances of ZnO and Zn_2_SnO_4_ NW sensors to NO_2_ and other reducing gases are displayed in Figure 3F. Zn_2_SnO_4_ NW exhibited significantly better response towards NO_2_ gas in comparison to ZnO NW. Also, ZS2 showed higher response than ZS4 and ZS6, because placing the sensors far from the source resulted in several surface defects due to the lack of a Sn source. Responses for Zn_2_SnO_4_ NW sensors with growth times of 15, 30 and 60 min are shown in Figure 3G. It is revealed that comparatively high or low density of NWs decreases the gas response.

### 2.4. GaN-Based NO_2_ Sensors

Having a wide bandgap energy (3.4 eV), gallium nitride (GaN) is found to support higher peak internal electric fields than silicon or gallium arsenide (GaAs). This wide bandgap causes lower thermal electron-hole pair generation, hence allowing high working temperatures. GaN is less vulnerable to attack in caustic environments, and resistant to radiation because of the larger cohesion energies among its constituent atoms [90,91,92,93]. Bishop et al. proposed a double Schottky junction gas sensor based on BGaN/GaN [94]. Two devices were developed; first, 10 periods of 20 nm thick undoped GaN, and 20 nm thick BGaN formed the BGaN/GaN superlattice structure. Then a circular diode having 300 µm as diameter was made with a 200 µm spacing between two Pt contacts on the n-type GaN sample (Figure 4A). When the sensors were exposed to 450 ppm NO_2_ gas at different temperatures, BGaN/GaN SL sensor exhibited higher current change and sensitivity than GaN monolayer sensors (Figure 4B). This enhancement is caused by two main reasons: firstly, BGaN has more interface traps than GaN, which creates more adsorption sites at the interface for gas molecules resulting a greater SBH change. Secondly, BGaN shows columnar growth thus a decrease in the volume-to-area ratio at the interface that provides more interface traps within a given area. It was found that at higher temperatures and concentrations, saturation of the signal change leads to a nonlinear response for the BGaN/GaN SL resulting into a decrease in the responsivity of the device (Figure 4C). In another study, an AlGaN/GaN high electron mobility transistor (HEMT) with Pt functionalized gate demonstrated a high sensitivity of 38.5% toward 900 ppm NO_2_ at high operating temperature of 600 °C [95]. The fabricated heterostructure sensor exhibited robustness under severe environmental conditions with a very quick response time of 1 s. When sensors are integrated in chips, low power sensor operation is required. Lim et al. [96] made SnO_2_ sensitized AlGaN/GaN sensor operating at ultra-low power without using any heater. The fabricated sensor exhibited ppb level detection as well as fast response times. 

### 2.5. Organic Materials-Based NO_2_ Sensors

Conducting and semiconducting organic films are promising gas sensing materials due to their excellent ability of tuning the chemical and physical properties on exposure to gas molecules. Also, recognition groups can be integrated covalently on organic sensing materials in order to get high selectivity and response [97]. Organic field effect transistors (OFETs) and thin film transistors (TFTs) are two major forms of organic material-based sensors. Kumar et al. [98] synthesized an OFET to detect NO_2_ gas using gate bias as control unit. The active layer of the OFET was the polymer poly [N-90-heptadecanyl-2,7-carbazole-alt-5,5-(40,70-di-2-thienyl-20,10,30-benzothiadiazole] (PCDTBT). The electron removal of NO_2_ molecule from the p–type conducting polymer PCDTBT led to an increase of conductivity. The typical transfer and output characteristics of OFET sensor are shown in Figure 5A,B, respectively. From the attained transfer and output characteristics, the mobility (µ_sat_) and threshold voltage (V_th_) were obtained as 1.13 × 10^−4^ cm^2^ V^−1^ s^−1^ and −9 V. From the gas concentration test, as shown in Figure 5C, the response increases linearly up to 10 ppm of exposure and then the increasing trend drops at higher concentrations. This happens because most of the active adsorption sites of the active PCDTBT layer get populated by NO_2_ molecules. The response and recovery time of the sensor at 1 ppm of NO_2_ exposure were obtained as shown in the inset of Figure 5C. The selectivity of the sensor upon exposure 10 ppm of different toxic gases was studied. Figure 5D displays that the OFET sensor exhibits the highest selectivity towards NO_2_ gas. Although the H_2_S gas response was moderate, the recovery was incomplete. In another study, a 6,13-bis(triisopropylsilylethynyl)-pentacene (TIPS-pentacene) film-based NO_2_ sensor attained a sensitivity above 1000%/ppm along with quick response-recovery [99]. It was predicted that the high sensing performance is attributable to the effective charge transport on the top of low original carrier concentration. Huang et al. [100] fabricated TFTs using copper phthalocyanine (CuPc) for NO_2_ gas detection. The gate dielectric used here is a UV–ozone (UVO)-treated polymer. Figure 5E shows sensitivities of the TFT biased at *V*_D_ = *V*_G_ = −40 V toward different NO_2_ concentrations and UVO treatment times (t_UVO_). It is seen that the sensitivity enhances significantly for sensors with longer t_UVO_ at all NO_2_ gas concentrations because of UVO-derived hydroxylated species on the dielectric surface. Gas selectivity tests revealed that without UVO treatment of the dielectric, the sensors are not at all selective to NO_2_ gas. However, at t_UVO_ = 360 s, the sensitivity increased from 10% to almost 600% at a concentration of 20 ppm NO_2_ which is six times more sensitive than all other test gases (Figure 5F).

### 2.6. Hybrid Materials-Based NO_2_ Sensors

In most cases hybrid materials combine and exhibit the useful characteristics of their constituent materials to promote high sensing performances. For instances, both MoS_2_ and RGO show good conductivity changes upon adsorption of NO_2_ molecules, and thus a combination of MoS_2_ and RGO results in high performance NO_2_ gas sensors [101]. Recently, Wang et al. synthesized a MoS_2_ nanoparticles-incorporated RGO hybrid material for NO_2_ detection by a two-step wet-chemical method [102]. In the first step, from powdered MoS_2_ NPs were prepared by a modified liquid exfoliation method. Next, self-assembly of MoS_2_ NPs and GO nanosheets, and hydrothermal treatment provided MoS_2_-RGO hybrid nanosheets. A high magnification SEM image of MoS_2_-RGO hybrids, shown in Figure 6A, reveals the presence of NPs on the RGO surface. It was found that the response time and recovery time decrease with increasing operating temperature while the sensor responses to NO_2_ remain almost unchanged. The optimum operating temperature was obtained as 160 °C. A response-recovery curve to 3 ppm NO_2_ gas at 160 °C is illustrated in Figure 6B. When the fabricated MoS_2_-RGO based sensor was exposed to NO_2_ gas concentrations ranging from 100 ppb to 3 ppm, the response followed an increasing trend due to the increased amount of NO_2_ molecules absorbed (Figure 6C). Wang et al. [103] synthesized a hybrid sensing material made of ZnO and poly(3-hexylthiophene) for the detection of NO_2_ at room temperature. The fabricated nanosheet-nanorod structured bilayer film sensor showed a sensitivity of 180% at 50 ppm of gas exposure. 

The sensing performance metrics like sensitivity/response, response and recovery times at certain gas concentration and operating temperatures, and sensitivity per ppm/response time ratio for different NO_2_ sensor materials and structures are summarized in Table 1. It provides a brief comparative performances outline among different NO_2_ sensor reported in recent years.

## 3. Recent Advances in SO_2_ Gas Detection

### 3.1. Carbon Material-Based SO_2_ Sensors

Aligned carbon nanotubes possess high surface-to-volume ratios which promote efficient physical and chemical adsorption of target gases [114]. Recently Zouaghi et al. have initiated research on vertically aligned carbon nanotube (VACNT)-based gas sensors interrogated by THz radiation [115]. They synthesized VACNT on SiO_2_ coated, boron-doped Si substrate by a water-assisted chemical vapor deposition method. Figure 7A shows a SEM image of vertically aligned CNT indicating a layer thickness of 95μm. The transmission spectrum upon SO_2_ gas exposure is illustrated in Figure 7B. The denser rotational spectrum of SO_2_ is attributed to the bent structure of SO_2_ molecule. The highest relative transmittance was obtained around 0.2 THz. When SO_2_ gas was flowed abruptly into a Si/SiO_2_/VACNT sensor, the maximum of transmitted electric field amplitude decreased to a steady value with fast response time of 2–3 min (Figure 7C). However, the recovery time was too long (>70 min). It has been predicted that the slow recovery was caused from the high sticking coefficient of SO_2_ gas to steel walls in the system. In a previous research, cholesteric-nematic mixture intercalated with CNT walls had been prepared and physical adsorption between the CNT and SO_2_ molecules was observed [116]. This adsorption phenomenon altered the CNT conductivity that in turn resulted into sensing signal for SO_2_. Zhang et al. synthesized TiO_2_/graphene film using layer-by-layer self-assembly technique for room temperature SO_2_ detection [117]. Excellent contacts between TiO_2_ and rGO are achieved from the wrapping of rGO flakes on TiO_2_ nanosphere surface or bridge-connection between TiO_2_ balls as shown in SEM image (Figure 7D). The sensor was exposed to 1, 50, 250, 1000 ppb SO_2_ gas to study the response-recovery behavior plotted in Figure 7E. It was observed that with increasing gas concentration, the sensor response kept increasing but the response-recovery time became longer. It has been predicted that the large interspace is responsible for the increase of response and recovery time. The TiO_2_/rGO film sensor showed much higher sensitivity to 1 ppm SO_2_ gas at room temperature than other target gases such as-—CH_4_, C_2_H_2_, H_2_, CO, NO_2_ (Figure 7F). So, the synthesized sensor was selective enough to SO_2_ gas. 

### 3.2. Metal and Metal-Oxide Nanostructures-Based SO_2_ Sensors

Many attempts had been made for SO_2_ gas detection using various semi-conducting metal oxides, such as- CeO_2_, WO_3_, V_2_O_5_-TiO_2_, MoO_3_-SnO_2_ and NiO [118]. However, due to instability in the highly reducing atmospheres, these sensors can only operate at low temperature (<500 °C) [119]. Liu et al. fabricated ZnO nanosheets decorated with Ru/Al_2_O_3_ catalyst and integrated them with a microsensor to detect SO_2_ gas [120]. Inkjet printing technology was used to load the sensor. AFM image in Figure 8A reveals the uniformity of the prepared ZnO 2D nanosheet and the thickness is indicated as about 1.5 nm. Different concentrations of SO_2_ gas had been exposed to Ru/Al_2_O_3_/ZnO sensor and the corresponding resistance responses are shown in Figure 8B. It is seen that resistance notably decreased at SO_2_ exposure and percent sensor response increased linearly with SO_2_ concentration. At 25 ppm of SO_2_, the obtained response and recovery times were about 1 min and 6 min, respectively. The SO_2_ selectivity test is displayed in Figure 8C, where the fabricated sensor responded negligibly to the test gases CO, CH_3_OH, C_2_H_5_OH, acetone, CO_2_, NO and HCHO in comparison to SO_2_ gas. From on-line mass spectrometry experiments, it was found that the catalyst Ru/Al_2_O_3_ dissociates SO_2_ molecules into easily detectable SO• species. Being captured by ZnO nanosheet, these species contribute to the sensor output signal. In another study, Ciftyürek et al. prepared and then evaluated molybdenum and tungsten binary and ternary oxide thick films for gas sulfur species sensing [121]. It was found that hydrothermally synthesized nano-SrMoO_4_ exhibited the highest sensor response among those fabricated oxide films. The SrMoO_4_-based sensors were able to operate at very high temperature (>600 °C) while maintaining their sensing performances, and thus can be useful in gas monitoring at industries. SnO_2_ thin film had been prepared by Tyagi et al. [122] using sputtering technique. Then, the film was functionalized with various metal oxide catalyst such as- PdO, CuO, NiO, MgO, V_2_O_5_ to make SO_2_ gas sensor. The uniform distribution of NiO nanoclusters on the surface of SnO_2_ film is noticed in the SEM image (Figure 8D). 500 ppm of SO_2_ gas was exposed to different metal-oxides deposited on SnO_2_ sensors to study the response at various operating temperatures (Figure 8E). NiO/SnO_2_ structure showed the highest response (∼56) at 180 °C due to two main reasons. Firstly, the spill-over effect from NiO nanoclusters toward SO_2_ molecules. Secondly, increase of adsorbed oxygen species sites at the porous and rough surface of SnO_2_ film [123]. The response and recovery time of NiO/SnO_2_ sensor were estimated as 80 s and 70 s respectively towards 500 ppm of SO_2_ gas at 180 °C as shown in Figure 8F. Also, the sensor exhibited good reproducibility and selectivity under SO_2_ exposure. In another study, it had been reported that BiFeO_3_ is highly selective to SO_2_ against carbon monoxide and butane [124]. Also, it was found that BiFeO_3_ synthesized by a sonochemical method provides better sensing performances than when prepared by conventional methods.

### 3.3. GaN-Based SO_2_ Sensors

AlGaN/GaN heterostructure semiconductors facilitate low power consumption, miniaturization and excellent sensing performances [125]. Also, AlGaN/GaN-based sensors can operate in chemically harsh environments, at high temperatures and under radiation fluxes due to having thermally and chemically stable structures [126]. Triet et al. synthesized Al_0.27_Ga_0.73_N/GaN-based Schottky diode sensors for SO_2_ gas detection [127]. Vertical zinc oxide nanorods (ZnO NRs) and a RGO nanosheet hybrid was formed on a AlGaN/GaN/sapphire heterostructure where the RGO and AlGaN surface made a Schottky contact with each other. From the FE-SEM image in Figure 9A, it is observed that neighboring ZnO NRs are attached to each other by RGO. During the gas exposure, the Schottky barrier between RGO and AlGaN layers changes. As a result, thermionic emission carrier transport is altered which in turn modifies the reverse saturation current. In the case of detecting SO_2_ (Figure 9B), the resistance response increased with increasing gas concentration because SO_2_ molecules are electron withdrawers. The non-linearity of the response with gas concentration is attributed to incomplete recovery of the sensing material RGO-ZnO NRs (Figure 9C). Here, SO_2_ gas molecules react with interaction sites resulting into slow diffusion of gas molecules within the RGO multilayer structure [128].

### 3.4. Solid Electrolyte-Based SO_2_ Sensors

Different solid electrolytes such as- NASICON [129], YSZ [130] and alkali metal sulfates [131] have been exploited during the past decades to fabricate high performance SO_2_ sensors. Among all the solid electrolytes, NASICON is widely used in the mixed-potential sensors due to its high ionic conductivity. Ma et al. [132] reported a mixed-potential gas sensor using NASICON and orthoferrite (La_0.5_Sm_0.5_FeO_3_) as sensing electrode. The SEM image of powdered La_0.5_Sm_0.5_FeO_3_ having a perovskite crystal structure reveals the uniformity of size and porosity (Figure 10A). La^3+^ doping level had been varied to study the variation of sensing performances. The highest response (−86.5 mV) was obtained for sensor with La_0.5_Sm_0.5_FeO_3_ as sensing electrode to 1 ppm SO_2_ (Figure 10B). The response order was found as ΔV(La_0.5_) > ΔV(La_0.4_) > ΔV(La_0.6_) > ΔV(La_0.8_) > ΔV (La_0.2_). The porous structure and electrocatalytic property are possibly responsible for the variation of responses. Responses were recorded at different operating temperatures. The equity between the amount of adhering gas and the activation energy demand indicated 275 °C as the optimum operating temperature with the highest response. The prepared mixed-potential sensor was exposed to other test gases such as- NO_2_, Cl_2_, NH_3_, CO, NO, acetone, H_2_, CH_4_, ethanol and methanol for a gas selectivity test. The sensor remained selective enough to detect SO_2_ gas even in very low amounts as illustrated in Figure 10C. In another study, a zirconia-based solid state electrochemical SO_2_ sensor had been demonstrated with MnNb_2_O_6_ as sensing electrode [133]. Under very high operating temperature (700 °C), the sensor attained good sensitivity along with rapid and stable response-recovery of gas molecules.

### 3.5. Zeolite-Based SO_2_ Sensors

Zeolites are aluminosilicates possessing immensely porous crystal structure, high specific surface area, high chemical and thermal stability, good adsorption properties, alterable chemical composition, presence of mobile ions, ability to undergo ion-exchange process and variable hydrophobic or hydrophilic features [134,135]. These characteristics make zeolites very attractive for gas detection. Choeichom et al. studied the effects of zeolite type, cation type and Si/Al ratio within various zeolites when exposed to SO_2_ gas [136]. During the exposure to 4200 ppm SO_2_, pristine zeolites exhibited the different sensor responses plotted in Figure 10D. It was found that the relative response of each pristine zeolite type showed the following decreasing order: ZSM-5 > beta > 13X > Y > 4A > ferrierite > mordenite > 5A > 3A. The three key factors contributing to the variation of these zeolite responses are pore size, cation type and Si/Al ratio. It was observed that the relative response increases with increasing zeolite pore size, however, decreases with a too large pore size. Among the monovalent cation zeolites focused here, the NH_4_^+^ zeolite response was the highest because of formation of hydrogen bonds with more than one SO_2_ molecule. With decreasing of Si/Al ratio, the responses kept increasing. The combined effect of the above discussed factors contributed to NH_4_^+^ZSM-5 (23) achieving the highest relative response toward SO_2_ with 23 as Si/Al ratio and medium pore size. Recovery and repeatability assessments were performed by flowing 4200 ppm SO_2_ for four cycles as illustrated in Figure 10E. The sensor conductivity returned to its initial value after SO_2_ was removed and again produced the same response to SO_2_ in the subsequent cycles. These results indicate the complete recovery and strong repeatability of the zeolite sensor. The sensor responses of various ion-exchanged ZSM-5 (23) towards 4200 ppm SO_2_ had been investigated as well (Figure 10F). It was found that Al^3+^ZSM-5 (23) provides the highest relative response due to two key factors: firstly, the magnitude of the ion-dipole attraction increases with the increasing ionic charge. Al^3+^ having higher ionic charge than Mg^2+^ and Na^+^, promotes a higher degree of interaction with SO_2_ molecules which in turn results in a higher sensor response. Secondly, the higher electronegativity of Al^3+^ZSM-5 (23) governs the stronger cation-dipole interaction with SO_2_ and thus facilitates a higher sensor response. Recently, a zinc-doped zeolitic imidazolate framework (ZIF-67) attached with CNT has been reported as SO_2_ sensing material [137]. It provided notable sensing performances at room temperature due to its porous polyhedral structure of metal particles with numerous interlinked CNTs. Previously, conductive polymer/zeolite composite based SO_2_ detection had been studied [138]. It was observed that PEDOT-PSS/KY zeolite composite achieved the highest sensor response having gas adsorption–desorption dependence on the cation types of Y zeolite.

### 3.6. Paper-Based SO_2_ Sensors

Sensing materials incorporated onto paper offer color transition sensing with the eyes, whereby measurement systems and electric circuits are not needed [139]. Paper-based analytical devices (PADs) provide the advantages of ease of production, low cost, flexibility, efficient sample collection, and easy disposability [140]. Li et al. coupled headspace sampling (HS) with PAD in order to detect SO_2_ through surface-enhanced Raman scattering (SERS) [141]. Hybrids consisting of 4-mercapto-pyridine (Mpy)-modified gold nanorods (GNRs) and reduced graphene oxide (rGO) were prepared. Then along with anhydrous methanol and starch iodine complex, the rGO/MPy-GNRs hybrids were immobilized upon cellulose-based filter papers using a vacuum filtration method. This process promotes the formation of a dense blue colored film on the filter paper as shown in the SEM images (Figure 11A–C). Uniform cellulose fibers of 12.5 μm width adopt wrinkle-like structures because of the attachment with rGO (Figure 11B). On exposing the fabricated rGO/MPy-GNRs/SIC paper to SO_2_, the blue color faded within minutes as illustrated in Figure 11C. It was found that the intermolecular charge-transfer complex between starch and iodine produces a broad band at 600 nm as indicated by curve d of Figure 11D.

The IR spectrum of rGO/MPy-GNRs/SIC is displayed in Figure 11E indicating the modification in the response after SO_2_ exposure. Along with the distinct and typical peaks for MPy, SO_2_ adsorption introduces a new peak having increased intensity in the SERS spectra of rGO/MPy-GNRs/SIC as shown by curve e in Figure 11F. This additional peak occurs because SO_2_ possibly affects the bending vibration of pyridine, and the characteristic peaks of SO_2_-pyridine complex are reflected in the bands. Recently, an amino-functionalized luminescent MOF material (MOF-5-NH_2_) was incorporated onto test paper for portable SO_2_ sensing [142]. It was seen that the prepared luminescent paper got lightened upon SO_2_ gas exposure with high selectivity. Also, it detected as low as 0.05 ppm SO_2_ having reusability advantages. In another research work, a microfluidic paper-based integrated detection system had been reported to monitor SO_2_ concentrations using RGB color analysis software [143]. The sensing performance metrics like sensitivity/response, response and recovery times at certain gas concentration and operating temperatures, and sensitivity per ppm/response time ratio for different SO_2_ sensor materials and structures have been summarized in Table 2. It provides a brief comparative performances outline among different SO_2_ sensor reported in recent years.

## 4. Recent Advances in H_2_S Gas Detection

### 4.1. Carbon Material-Based H_2_S Sensors

In recent years, many research efforts have been made for H_2_S detection using graphene, reduced graphene oxide and carbon nanofibers. Ovsianytskyi et al. [152] proposed a graphene-based H_2_S gas sensor functionalized with Ag nanoparticles (Ag NPs) and charged impurities. Graphene was grown by the CVD technique, and then Ag NPs and impurities were incorporated on the graphene by a wet chemical method. The SEM image of graphene after immersing into AgNO_3_/Fe(NO_3_)_3_ solution reveals the presence of large number of nanoparticles (10–100 nm) uniformly distributed on its surface (Figure 12A). The comparative responses obtained on exposing 500 ppb of H_2_S gas for 400 s to pristine graphene, graphene doped with Fe(NO_3_)_3_ solution, graphene doped with AgNO_3_ solution, and graphene doped with a mixed AgNO_3_/Fe(NO_3_)_3_ solution are displayed in Figure 12B. Graphene doped with the mixed solution exhibited the highest response. Since Ag is less electronegative than graphene, adsorption of H_2_S occurs because of its interaction with the adsorbed oxygen species on Ag mostly. Then, electrons released from dissociation of H_2_S are accumulated in graphene. This phenomenon causes a decrease in graphene hole concentration, and thus resistance of Ag-doped graphene increases.

The relationship between gas concentrations and corresponding relative responses of the synthesized sensor is quite linear, as plotted in Figure 12C. Also, the sensor was strongly selective to H_2_S gas against CH_4_, CO_2_, N_2_, and O_2_ gases. Similarly, Chu et al. [153] obtained a sensitivity of 34.31% toward 100 ppm H_2_S with tin oxide-modified reduced graphene oxide (SnO_2_-rGO) at 125 °C. In another study, Zhang et al. [154] developed a stable sensor using ZnO-carbon nanofibers (30.34 wt% carbon) that exhibited good H_2_S sensing performances. It was found that the protection of carbon provides high stability of ZnO and oxygen vacancies to allow improved sensor responses.

### 4.2. GaN-Based H_2_S Sensors

Several AlGaN/GaN-based gas sensors have been reported, including NO, NO_2_, NH_3_, Cl_2_, CO, CO_2_ and CH_4_ [156,157,158]. However, H_2_S sensing using wide bandgap semiconductors like GaN have not been explored yet that much. In order to sense H_2_S gas even at very low amounts, Sokolovskij et al. [155] synthesized AlGaN/GaN HEMT-based sensor with platinum as gate. The top view optical micrograph of the synthesized device shown in Figure 12D reveals the gate dimensions, gate-source and gate-drain spacing. For high temperature operations, each device was wire bonded to ceramic substrates. The variation of drain current was observed under different H_2_S concentrations and gate bias voltages (Figure 12E).

Because of the increasing baseline current with increasing gate bias, variation of the drain current was highly influenced. Also, the fabricated HEMT sensor operates in a wide range of biasing conditions without degrading the sensing performances and thus shows an excellent stability. It was found that when gate bias approaches pinch-off state, it minimizes power consumption and thus enables the sensor to operate at high response mode. The sensitivity of AlGaN/GaN HEMT sensor clearly increases with higher temperatures as plotted in Figure 12F. The rise and fall time were estimated 219 s and 507 s, respectively, at 250 °C. At lower temperatures, rise and fall times have gone higher. Further, Zhang et al. [159] pre-treated the Pt-gated AlGaN/GaN HEMT sensor with H_2_ pulses in dry air ambient at 250 °C. This treatment facilitated the enlargement of the H_2_S detection range up to 90 ppm. 

### 4.3. Metal and Metal Oxide-Based H_2_S Sensors

#### 4.3.1. Nanostructured Metal Oxide-Based Sensors

It was found that metal oxides such as- SnO_2_, WO_3_, ZnO, and α-Fe_2_O_3_ based sensors exhibited superior sensing performances toward H_2_S due to their stable nanostructures [160,161,162]. Zhang et al. [163] reported a α-Fe_2_O_3_ nanosheet-based H_2_S gas sensor using a solvothermal method. Figure 13A shows the SEM image of a sample obtained at reaction temperature of 160 °C denoted as S_160_. It was observed that at low temperatures, the morphology of the samples is not uniform. Since both α-Fe_2_O_3_ and Fe_3_O_4_ exist in the nanostructure, uniform morphology can’t be obtained under the reaction temperature of 160 °C. It was seen that sensor response of α-Fe_2_O_3_ to H_2_S decreases with the increasing working temperature. However, the recovery time is too long at low temperature, so taking sensor response, response time and recovery time into account, 135 °C was estimated as the optimum working temperature. Figure 13B displays the response of the prepared sensor to different concentrations of H_2_S ranging from 1 to 50 ppm at 135 °C. The response and recovery time were estimated to be less than 10 s and 45 s, respectively, indicating very a rapid response in comparison to other H_2_S gas sensors. The changes in the electric resistances were found negligible for the sensor to 50 ppm acetone, ethanol, methanol and H_2_ gases at 135 °C. On the contrary, the sensor response was very large to H_2_S under same conditions, thus reflecting excellent selectivity of the α-Fe_2_O_3_ nanosheet-based H_2_S sensor (Figure 13C). 

In another study, Li et al. [164] developed ZnO/CuO nanotube arrays to sense H_2_S at low-working temperatures. It was observed that the nanotube structures promoted the diffusion and adsorption of gas with many active sites between H_2_S molecules and adsorbed oxygen molecules. Thus, they contributed to achieve good sensitivity along with fast response time. It was found that porous In_2_O_3_ nanoparticles provide large surface areas and pore volumes which create numerous active sites to produce active oxygen species [165]. These sites facilitate a significant improvement in H_2_S gas sensing with 1 ppb of detection limit. Also, a dense array of intrinsic ZnO NWs has been reported for H_2_S detection by exploiting a sulfuration–desulfuration reaction mechanism [166]. In another work, Eom et al. [167] fabricated Cu_x(x=1,2)_O:SnO_2_ thin films to detect H_2_S gas at room temperature. Enhanced sensitivity with rapid response-recovery was obtained due to enhanced adsorption sites arising from abounding domains of p-n heterojunctions on the Cu_x_O:SnO_2_ film surfaces. Besides, a Cu-doped BaSrTiO_3_-based H_2_S sensor was reported [168]. Herein along with gas-surface interaction, the role of pre-adsorbed oxygen species and surface dipolar hydroxyl groups has been investigated as well. 

#### 4.3.2. Mesoporous Metal Oxide-Based Sensors

Generally mesoporous materials contain pores with diameters of 2–50 nm. Mesoporous metal-oxides offer efficient gas detection because they have large surface areas, open porosity, small pore sizes, and the ability to coat the surface of the mesoporous structure with one or more compounds. Quang et al. [169] reported a mesoporous Co_3_O_4_ nanochains-based H_2_S sensor. At first cobalt carbonate hydroxide (Co (CO_3_)_0.5_(OH)·11H_2_O) nanowires were synthesized using a hydrothermal route. Then heat treatment was applied in air at 600 °C for 5 h to form rough-surfaced mesoporous Co_3_O_4_ nanochains as shown by the TEM image in Figure 13D. From the gas responses vs. working temperatures analysis, the optimum working temperature was obtained as 300 °C. At lower operating temperatures than the optimum, Co_3_O_4_ nanochains displayed sluggish chemical activity causing weak responses. Moreover, at higher working temperatures, adsorbed H_2_S molecules start escaping from the Co_3_O_4_ nanochain surface because of increased activation. As a result, the sensor response starts decreasing. The fabricated sensor exhibited quick responses and recovery to 1–100 ppm H_2_S at 300 °C. At 100 ppm H_2_S, response and recovery times were estimated as 46 s and 24 s, respectively, as illustrated in Figure 13E. The nanochain structure provides a high specific surface area, narrow pore size and rich mesopores which make the fabricated sensor more suitable for H_2_S sensing than other toxic gases. The comparative responses of Co_3_O_4_ nanochains toward H_2_S and other target gases are plotted in Figure 13F indicate strong selectivity for H_2_S. Previously, Stanoiu et al. [170] prepared a mesoporous SnO_2_-CuWO_4_-based cost-effective H_2_S sensor having high sensitivity at low working temperature. In addition, a short temperature trigger of 500 °C was applied to enhance the recovery operation of the fabricated sensor.

#### 4.3.3. Metal Oxide Microsphere-Based Sensors

Typically, microspheres are small spherical particles having diameters in the micrometer range. Hu et al. [171] reported CuFe_2_O_4_ nanoparticles-decorated CuO microspheres-based H_2_S gas sensors. The synthesized CuO/CuFe_2_O_4_ heterostructures provided a porous and rough surface due to the arbitrary deposition of nanoparticles as displayed by the FE-SEM image in Figure 14A. When the temperature is low, the response becomes low because of the weak chemical interaction between the gas molecules and adsorbed oxygen species. At higher working temperatures, the mentioned chemical interaction is strong and the responses keep increasing, but gas molecule diffusion becomes slower than the surface interaction causing a decrease of the response again, as illustrated in Figure 14B. The optimal operating temperature was estimated as 240 °C. The dependence of responses of the sensor on H_2_S concentrations is plotted in Figure 14C which exhibits a gradual increasing trend. The response and recovery time of the fabricated CuO/CuFe_2_O_4_ sensor were obtained as 31 s and 40 s, respectively, at the optimal operating temperature (240 °C) with good reproducibility and selectivity toward H_2_S gas. In another study, Li et al. [172] prepared SiO_2_@TiO_2_ microspheres and then formed Cd^2+^-doped TiO_2_ shell-modified ITO electrodes for H_2_S detection. Exploiting the mismatch of energy band levels between TiO_2_ shells and induced CdS nanoparticles, this device provided good sensing performances.

### 4.4. MOF-Based H_2_S Sensors

Metal organic frameworks (MOFs) offer highly selective and sensitive detection of H_2_S because of possessing chemical stability, custom tuning of porosity and functionalities, and various pre- or post-synthetic modifications to the structural framework [173]. Guo et al. [174] synthesized a MOF material named as Zr(TBAPy)_5_(TCPP) using a solvothermal method, where Zr is the metal center, and 1,3,6,8-tetra(4-carboxylphenyl) pyrene (TBAPy) and tetrakis(4-carboxyphenyl) porphyrin (TCPP) act as double linkers. The prepared Zr(TBAPy)_5_(TCPP) exhibited well-shaped shuttle structures with a particle size of about 100 nm as seen from the transmission electron microscope (TEM) image (Figure 14D). However, Zr-MOF NU-1000 (synthesized for comparison) exhibited an irregular structure indicating the structural effect of TCPP on the synthesized materials. The FTIR spectrum of Zr(TBAPy)_5_(TCPP) is displayed in Figure 14E. There is a clear shift in the N-H and C=N peak on the addition of S^2−^ due to the attachment between S and N in the materials. Fluorescence enhancement of the fabricated Zr(TBAPy)_5_(TCPP) sensor provides a linear trend with the increase of S^2−^ concentration as plotted in Figure 14F. The interference effects of other anions such as- SO_4_^2−^, CNS^−^, COOH^−^, Br^−^, I^−^, IO3^−^, F^−^, HSO_3_^−^, Cl^−^ and NO_3_^−^ was investigated and the results confirm that Zr(TBAPy)_5_(TCPP) is highly selective for S^2−^ sensing. In another study, Dong et al. [175] developed a ZIF-67-derived porous dodecahedra Co_3_O_4_ sensor showing enhanced linear trend with H_2_S concentration. The significant improvement in the overall sensing performances is attributed to a high specific surface and the exposed {110} lattice planes of the fabricated structure.

### 4.5. Organic Materials-Based H_2_S Sensors

In recent years, several attempts have been made for H_2_S sensing using organic semiconducting films and polymers. Different types of interactions like crosslinking, doping, grafting and scissioning between electrons and organic materials take place in subject to energy as well as the dose of incident electron beam [176] which help to attain high selectivity and sensitivity to target gas. Chaudhary et al. reported a polyaniline-silver (PANI-Ag) nanocomposite film-based H_2_S sensor [177]. After protonation of aniline monomers, photopolymerization of aniline on a bi-axially oriented polyethylene terephthalate (BOPET) sheet was performed. The prepared PANI-Ag films were irradiated by a 10 MeV electron beam. As the dose was increased, the nanofiber diameter increased and a 30 kGy dose promoted an interconnected microstructure with larger sized Ag particles. At a very high dose of 100 kGy, Ag clusters submerged inside the polymer matrix with denser structure. The bright spots observed in SEM image as shown in Figure 15A reveal that Ag nanoparticles are incorporated in the PANI matrix. After EB irradiation, Ohmic nature was retained as seen from the linear I-V relationship displayed in Figure 15B. The electrical conductivity kept increasing with the dose to achieve the highest value at 30 kGy and then started going down. The percent sensor responses under different H_2_S concentrations and irradiation doses are plotted in Figure 15C. Since lower irradiation doses cause higher conductivity changes, the corresponding sensing responses become lower upon H_2_S exposure. On the contrary, higher doses cause lower electrical conductivity due to crosslinking-induced structural defects, so, the corresponding sensing responses to H_2_S become larger. Abu-Hani et al. [178] engineered the conductivity of chitosan (CS) film to obtain a highly sensitive and selective sensor toward H_2_S gas. Glycerol ionic liquid (IL) had been incorporated to tune the conductivity of the CS film. It was able to operate at lower temperature and provided rapid response-recovery with low-power consumption. In another study, Cu^2+^-doped SnO_2_ nanograin/polypyrrole nanospheres-based H_2_S gas detection was reported [179]. The enhanced sensing performance of the fabricated organic-inorganic nanohybrids is mainly attributed to improved surface potential barrier by surface defects tailoring, and numerous reaction sites to accelerate gas diffusion and adsorption. 

### 4.6. Solid Electrolytes-Based H_2_S Sensors

The mixed potential type sensor requires a solid electrolyte having the ability to transfer oxygen ions between reference electrode and sensing electrode. Hao et al. [180] prepared a mixed-potential H_2_S gas sensor using YSZ as solid electrolyte and La_2_NiO_4_ as sensing electrode. The microstructure of the sensing material La_2_NiO_4_ was varied with three equivalents of citric acid and total metal irons which were 0.5:1 (LNO-0.5), 1:1 (LNO-1) and 1:2 (LNO-2), respectively. The SEM image of LNO-1 sensing material (La_2_NiO_4_ powders) reveals the porous structure and it has the largest pore size among the three samples as displayed in Figure 15D. Also, it was found that LNO-1 possesses the highest BET surface area and pore volume. On increasing the H_2_S exposure concentration, the electrode potential difference exhibited linear changes with the logarithm of H_2_S concentration at 500 °C as plotted in Figure 15E. The sensitivity of the sensor to H_2_S was changed to −10 mV/decade from −69 mV/decade. The recovery time was improved by applying a temperature pulse of 700 °C. For 500 ppb of H_2_S exposure, it decreased from 20 min to 150 s. The fabricated sensor was proved to be highly selective toward H_2_S compared to other target gases as observed from the responses toward 1 and 2 ppm of every test gas (Figure 15F). Moreover, it was found quite stable in long term performance with lower detection limit. In another study, Yang et al. [181] used Nafion as a proton exchange membrane to demonstrate H_2_S sensing with the help of a sensing electrode. The electrode was made of Pt-Rh nanoparticles loaded on carbon fibers. The sensitivity was obtained 0.191 μA/ppm from the linear plot between sensor current changes and corresponding H_2_S concentrations. The fabricated sensor was highly selective toward H_2_S at room temperature with a fast recovery time (16 s) under 50 ppm of gas exposure. Recently, a promising TMD material, WS_2_ has been utilized to detect H_2_S with high sensitivity and selectivity [182]. It was observed that oxygen doping in the sulfur sites of the WS_2_ lattice promotes enhanced sensing performances towards H_2_S. Earlier, the adsorption properties of WS_2_ had been analyzed toward various target gas molecules along with its Fermi level pinning [183]. The sensing performance metrics like sensitivity/response, response and recovery times at certain gas concentration and operating temperatures, and sensitivity per ppm/response time ratio for different H_2_S sensor materials and structures have been summarized in Table 3. It provides a brief comparative performances outline among different H_2_S sensors reported in recent years.

A novel metric, sensitivity per ppm/response time ratio has been calculated for each sensor in order to compare the overall sensing performance on the same reference. The higher value of the calculated ratio indicates the better overall sensor performance. Average ratios have been obtained by taking the recently reported gas sensors into account for the highly focused sensing materials as illustrated in Figure 16. It is found that hybrid materials-based sensors exhibit the highest average ratio for NO_2_ gas sensing, whereas GaN and Metal-oxide based sensors possess the highest ratio for SO_2_ and H_2_S gas sensing respectively.

## 5. Recent Density-Functional Theory (DFT) Study of Gas Molecule-Sensor Interaction

Numerous efforts have been made to investigate the adsorption properties of various sensing materials toward different toxic gases including NO_2,_ SO_2_ and H_2_S by first-principle method calculations using density functional theory (DFT) in recent years as shown in Table 4. 

For instance, Chen et al. [202] reported that NO_2_ and SO_2_ adsorptions show chemisorption character on boron-doped arsenene whereas physisorption character on pristine and nitrogen-doped arsenene. Mao et al. [203] explored the effect of Ge/Se vacancy, anti-site defect, and P atom substituted defect on GeSe monolayer toward toxic gas adsorption. It was found that, the point defect engineering alters electronic structure and work function of GeSe monolayer and thus influence on the adsorption properties of target gas molecules. Besides, adsorption properties such as- adsorption energy, shortest adsorption distance, charge transfer estimation, stability etc. toward NO_2,_ SO_2_ and H_2_S have been studied based on various sensing materials like borophene [204], monolayer C_3_N [205], blue phosphorene [206], organolithium (C_2_H_4_Li) complex [207], Ni-MoS_2_ monolayer [208], germanene nanosheet (Ge-NS) [209], Fe-atom-functionalized CNTs [210], 2D tetragonal GaN [211], graphitic GaN sheet [212], etc.

## 6. Calibration of Toxic Gas Sensors

In order to check sensor precision, toxic gas sensors must be calibrated at regular intervals. The sensor producers typically suggest a time interval between calibrations. Single toxic gas detectors are normally calibrated with a defined toxic gas depending on the gas type whereas multi-gas detectors are calibrated with their own specific calibration gas mixtures. There are mainly two steps in the gas sensor calibration. Firstly, a reference zero reading must be established using pure nitrogen or pure synthetic air. Secondly, the sensor operating range must be calibrated using a standard gas mixture. The ideal practice is to apply a mixture of the target gas balanced in the natural air as the calibration gas. Premixed calibration gas, permeation devices, cross calibration, gas mixing, Gaussian processes are some of the practical methods of calibrating the gas sensors [213,214,215]. 

## 7. Toxic Gas Sensors in Internet of Things (IoT) Applications

The Internet of Things is a network of physical objects that utilizes sensors and application programming interfaces (APIs) to collect and exchange data over the internet. IoT network requires ultra-low power, low cost, long lifetime, integrable into electronic circuits, and mini-sized gas sensors for remote air quality monitoring and enhanced automated system [216]. Electrochemical gas sensors can provide these characteristics required by IoT platforms, thus become suitable candidate for the IoT applications such as creating smart environment, smart home, smart parking system and so on [217,218]. Toxic gas sensors were incorporated into a multi-purpose field surveillance robot which uses multiple IoT cloud servers [219]. High performance gas sensors are utilized in IoT-based vehicle emission monitoring systems [220]. Besides, wireless sensor networks have been employed for toxic gas boundary area detection in large-scale petrochemical plants [221]. However, the gas sensing performances are strongly affected by miniaturization of sensor in terms of length and width between the electrodes, number of electrodes, sensing area etc. [222]. Extensive studies on sensing properties of miniaturized gas sensors can further facilitate the implementation of toxic gas sensors in IoT platforms.

## 8. Future Perspectives and Conclusions

Toxic gas sensors play an important role in many aspects of technology, industry, or daily life. In recent years, researchers have exploited the fundamental properties of various gas sensing materials to achieve high performance toxic gas sensors. Particularly, excellent improvements have been attained in terms of sensitivity, selectivity, limit of detection, miniaturization and portability for NO_2_, SO_2_ and H_2_S gas sensors using novel combination of nanomaterials exhibiting various morphologies. However, the toxic gas sensors reported so far have limitations in some of the important performance metrics, such as- response and recovery times, stability, operating temperature, reproducibility, fabrication cost, reliability etc. These limitations can be overcome by further exploiting the hybrid and heterostructure, exploring more in surface functionalization, and adopting novel, efficient and cost-effective fabrication technique. This work reviews and categorizes the recent progress in electrochemical detection of NO_2_, SO_2_ and H_2_S gases based on various highly explored sensing materials over the past few decades. Moreover, the sensing performance parameters like sensitivity/ response, response and recovery times at certain gas exposure concentration and operating temperature for various sensor materials and structures have been tabulated which provide a brief comparative performances outline to the reader. This study will give an overview on the research trend of the above-mentioned toxic gas sensors to the current and future researchers.

## Figures and Tables

**Figure 1 sensors-19-00905-f001:**
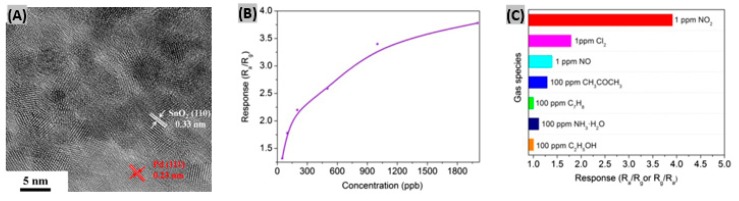
(**A**) HR-TEM image of fabricated Pd-SnO_2_-RGO hybrid. (**B**) Sensor response toward different concentrations of NO_2_ gas at room temperature. (**C**) The response of the sensor to Cl_2_, NO, NO_2_, acetone, toluene, ammonia and ethanol in a selectivity test. Figures adapted with permission from [67], Copyright 2018 Elsevier.

**Figure 2 sensors-19-00905-f002:**
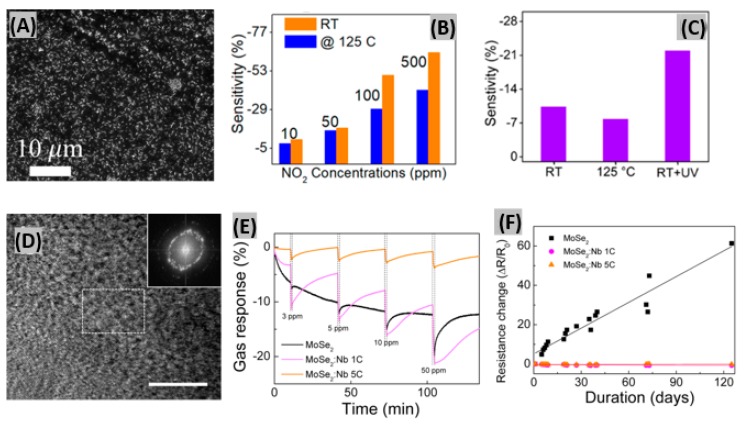
(**A**) FE-SEM image of the mixed MoS_2_ flakes. (**B**) Sensitivity vs. NO_2_ concentration bar graph at RT and 125 °C. (**C**) Sensitivity bar graph of the mixed MoS_2_ flakes-based sensor for a NO_2_ gas concentration of 10 ppm. Figures adapted with permission from [76], Copyright 2018 American Chemical Society. (**D**) Planar ADF-STEM image of the MoSe_2_:Nb 1C film (white scale bar = 10 nm). Inset shows the corresponding FFT patterns. (**E**) Percent gas response for MoSe_2_, MoSe_2_:Nb 1C, and MoSe_2_:Nb 5C sensors at 3 to 50 ppm of NO_2_ gas. (**F**) Response of the MoSe_2_, MoSe_2_:Nb 1C, and MoSe_2_:Nb 5C sensors over 120 days. Figures adapted with permission from [78], Copyright 2017 American Chemical Society.

**Figure 3 sensors-19-00905-f003:**
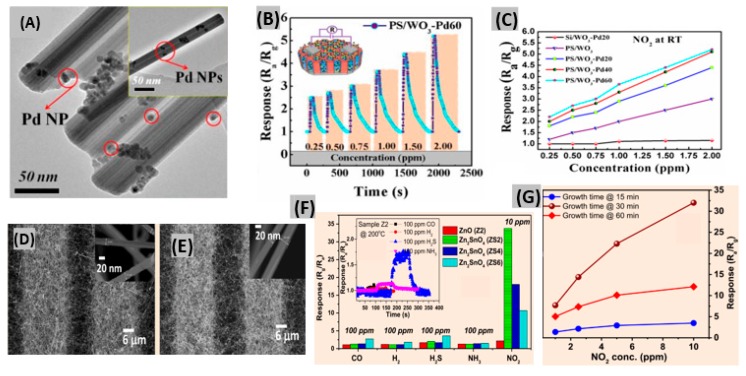
(**A**) TEM images of PS/WO_3_–Pd60 (inset shows a single PS/WO_3_–Pd60 NR). (**B**) Sensor response of PS/WO_3_–Pd60 to different concentrations of NO_2_ at RT. (**C**) Response of Si/WO_3_–Pd20, PS/WO_3_, PS/WO_3_–Pd20, PS/WO_3_–Pd40 and PS/WO_3_–Pd60 sensors at different NO_2_ concentrations. Figures adapted with permission from [84], Copyright 2018 MDPI. (**D**,**E**) FESEM images of on-chip grown ZnO and Zn_2_SnO_4_ NWs. (**F**) Response comparison of Z2, ZS2, ZS4, ZS6 to 100 ppm CO, H_2_, H_2_S, NH_3_ and 10 ppm NO_2._ (**G**) Sensor responses of the devices with growth times of 15 min, 30 min and 60 min as a function of NO_2_ concentrations. Figures adapted with permission from [89], Copyright 2018 Elsevier.

**Figure 4 sensors-19-00905-f004:**
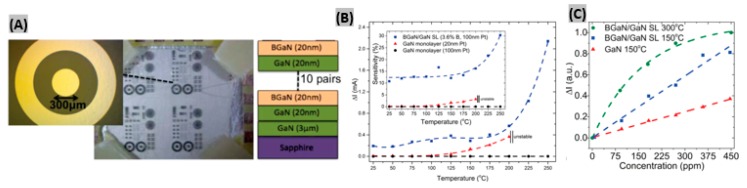
(**A**) Fabricated BGaN/GaN SL and GaN devices. (**B**) Current change and Sensitivity (Inset) of BGaN/GaN SL and GaN sensors to 450 ppm NO_2_ at 5 V bias at different temperatures. (**C**) Sensor current changes vs. NO_2_ concentrations for various temperatures. Figures adapted with permission from [94], Copyright 2015 AIP Publishing LLC.

**Figure 5 sensors-19-00905-f005:**
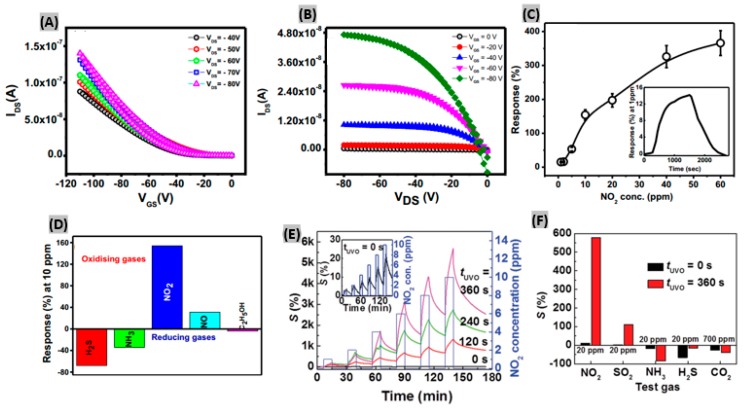
(**A**) Transfer (**B**) Output characteristics of the PCDTBT based OFET. (**C**) Sensor response at different NO_2_ concentrations. Response-recovery of the sensor at 1 ppm of NO_2_ (inset). (**D**) Selectivity graph of the sensor towards various analytes. Figures adapted with permission from [98], Copyright 2018 Elsevier. (**E**) Sensitivity profile at different NO_2_ concentrations and with UVO treatment times (t_UVO_). Inset shows the sensitivity at t_UVO_ = 0 s. (**F**) Sensitivities obtained for TFT sensors with t_UVO_ = 0 and 360 s at 20 ppm NO_2_, SO_2_, NH_3_, H_2_S and 700 ppm CO_2_. Figures adapted with permission from [100], Copyright 2017 WILEY-VCH Verlag GmbH & Co. KGaA, Weinheim.

**Figure 6 sensors-19-00905-f006:**
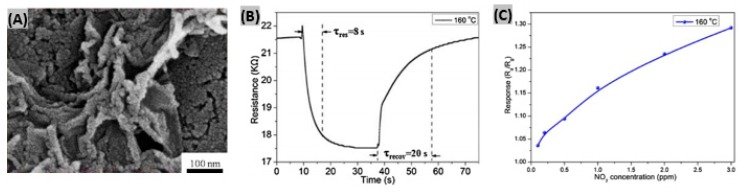
(**A**) High magnification SEM images of the fabricated MoS_2_-RGO hybrids. (**B**) Response- recovery curve of MoS_2_-RGO sensor to 3 ppm NO_2_ at 160 °C. (**C**) Sensor responses to different NO_2_ concentrations at 160 °C. Figures adapted with permission from [102], Copyright 2017 Elsevier.

**Figure 7 sensors-19-00905-f007:**
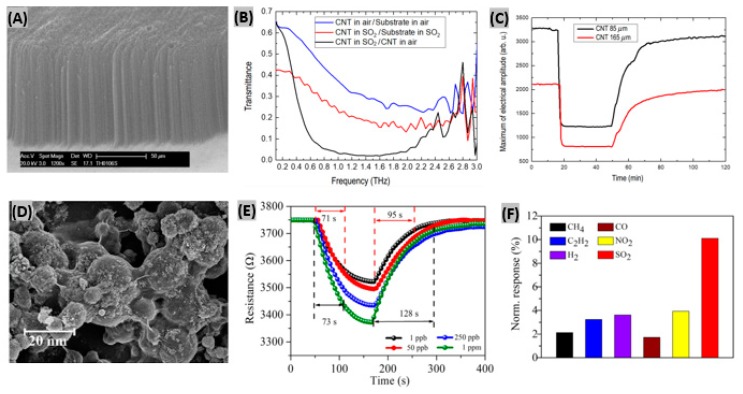
(**A**) SEM image of a VACNT layer on Si/SiO_2_ substrate. (**B**) Normalized transmittance spectra of Si/SiO_2_/VACNT in air and SO_2_. (**C**) Maximum of electrical amplitude vs. time response in transmission measurements with 85 um and 165 um thick layers of VACNT. Figures adapted with permission from [114], Copyright 2018 Lietuvos mokslų akademija. (**D**) SEM image of self-assembled TiO_2_/rGO film. (**E**) Response/recovery curves of TiO_2_/rGO sensor exposed to four different concentrations of SO_2_ gas. (**F**) Normalized response of TiO_2_/rGO sensor toward 1 ppm of various gases at room temperature. Figures adapted with permission from [116], Copyright 2017 Elsevier.

**Figure 8 sensors-19-00905-f008:**
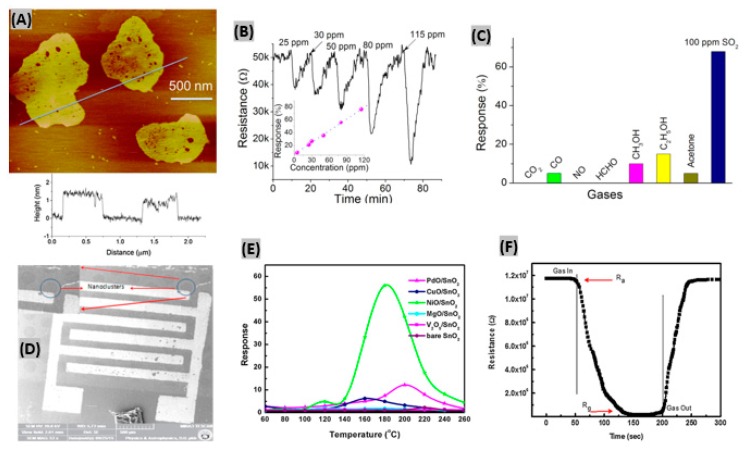
(**A**) Two-dimensional AFM image of ZnO nanosheets, height profile of two ZnO nanosheets as marked is shown below. (**B**) Resistance changes to SO_2_ gas at different concentrations, inset shows plotting of sensing response vs. SO_2_ concentration (**C**) Selectivity test of Ru/Al_2_O_3_/ZnO sensor to various gases under same concentration. Figures adapted with permission from [119], Copyright 2018 Elsevier. (**D**) SEM image of NiO/SnO_2_ sensor structure, inset displays the presence of NiO nanoclusters. (**E**) Sensor response of NiO/SnO_2_, PdO/SnO_2_, CuO/SnO_2_, MgO/SnO_2_, V_2_O_5_/SnO_2_ and bare SnO_2_ structures with operating temperatures towards 500 ppm of SO_2_ gas. (**F**) SO_2_ gas response-recovery illustration for NiO/SnO_2_ sensor. Figures adapted with permission from [121], Copyright 2015 Elsevier.

**Figure 9 sensors-19-00905-f009:**
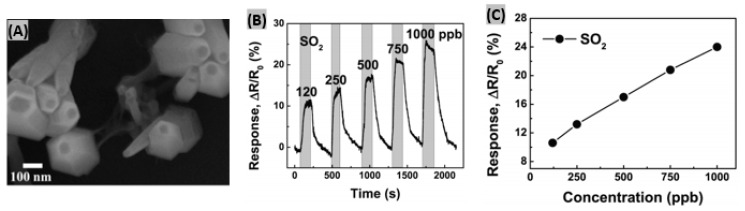
(**A**) Top-view of FE-SEM high magnification image of the RGO nanosheets connecting to ZnO NRs on AlGaN/GaN heterostructure. (**B**) Sensor resistance variations at the exposure to SO_2_ gas of different concentrations. (**C**) Response vs. gas concentration relationship under SO_2_ gas exposure. Figures adapted with permission from [126], Copyright 2017 American Chemical Society.

**Figure 10 sensors-19-00905-f010:**
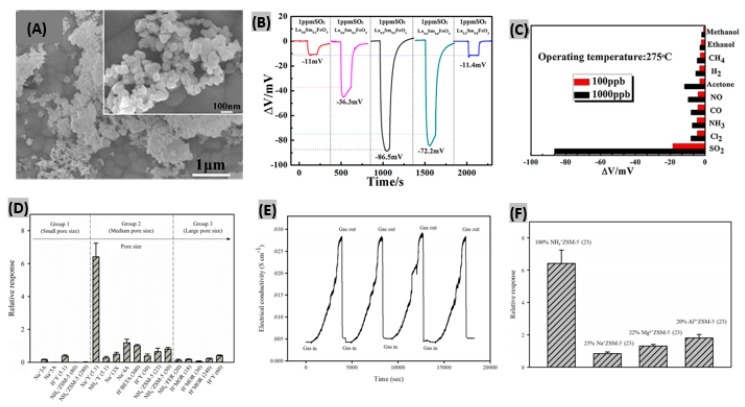
(**A**) SEM image of powdered La_0.5_Sm_0.5_FeO_3_, inset shows higher magnified image. (**B**) The voltage change response of the gas sensor made with La_x_Sm_1−x_FeO_3_ sensing electrodes (x = 0.2, 0.4, 0.5, 0.6, 0.8). (**C**) The selectivity test for the sensor using La_0.5_Sm_0.5_FeO_3_ as sensing electrode to various gases at 275 °C. Figures adapted with permission from [131], Copyright 2017 Elsevier. (**D**) Comparative responses of different pristine zeolites. (**E**) Sensor response of NH_4_^+^ZSM-5 (23) exposed to 4200 ppm SO_2_ for four cycles. (**F**) Comparative responses of different ion-exchanged ZSM-5 (23) zeolite sensors. Figures adapted with permission from [134], Copyright 2018 Springer-Verlag GmbH Germany, part of Springer Nature.

**Figure 11 sensors-19-00905-f011:**
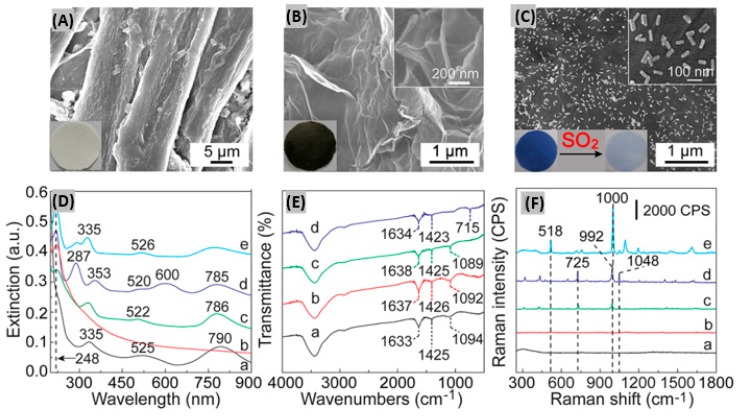
SEM images of (**A**) pure cellulose paper, (**B**) after assembling with rGO and (**C**) rGO/MPy-GNRs/SIC paper. Insets of figures (**A**–**C**) show pictures of the corresponding paper substrate under light. (**D**) UV-vis-NIR extinction spectra of (a) MPy-GNRs paper, (b) rGO paper, (c) rGO/MPy-GNRs paper, rGO/MPy-GNRs/SIC paper (d) before and (e) after the adsorption of SO_2_, respectively. (**E**) FT-IR spectra of (a) MPy-GNRs, (b) rGO/MPy-GNRs, rGO/MPy-GNRs/SIC (c) before and (d) after the adsorption of SO_2_, respectively. (**F**) SERS spectra of (a) the pure cellulose paper, (b) the rGO paper, (c) the MPy-GNRs paper, rGO/MPy-GNRs/SIC paper (d) before and (e) after adsorption of SO_2_, respectively. Figures adapted with permission from [140], Copyright 2018 American Chemical Society.

**Figure 12 sensors-19-00905-f012:**
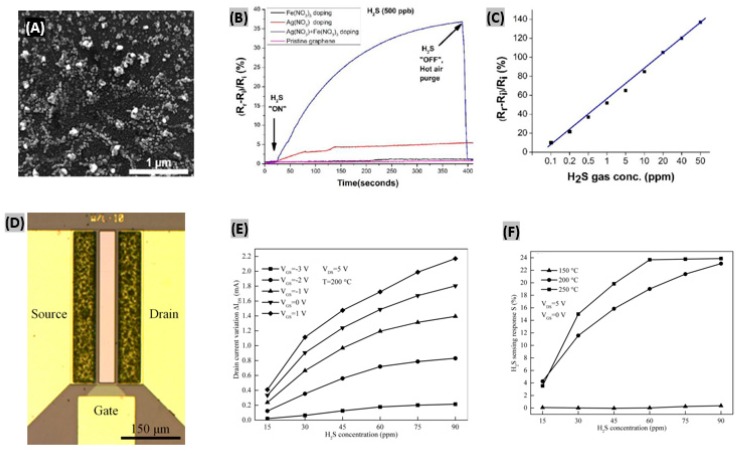
(**A**) SEM image of graphene after immersion into AgNO_3_/Fe(NO_3_)_3_ solution. (**B**) Comparative sensor responses of pristine graphene, graphene doped with Fe(NO_3_)_3_ solution, graphene doped with AgNO_3_ solution, and graphene doped with mixed Fe(NO_3_)_3_ and AgNO_3_ solution. (**C**) Relationship between H_2_S gas concentrations and corresponding relative responses of the fabricated sensor. Figures adapted with permission from [152], Copyright 2017 Elsevier. (**D**) Top view of the optical micrograph of the fabricated Pt-AlGaN/GaN HEMT sensor. (**E**) Changes in drain current with different concentrations of H_2_S gas and gate biases at T = 200 °C, VDS = 5 V. (F) Sensitivity of the fabricated sensor toward H_2_S at different temperatures. Figures adapted with permission from [155], Copyright 2018 Elsevier.

**Figure 13 sensors-19-00905-f013:**
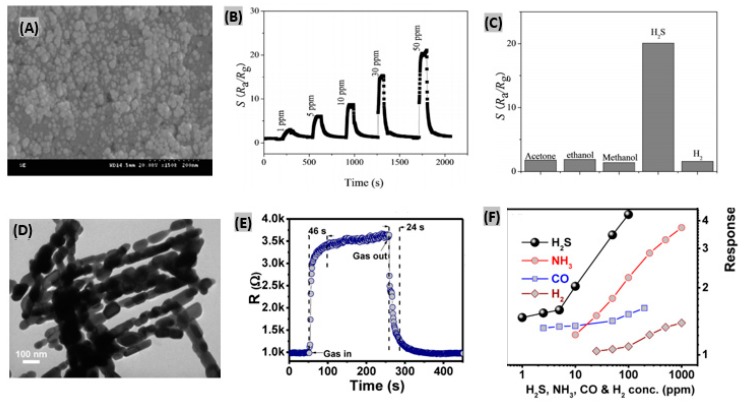
(**A**) SEM image of α-Fe_2_O_3_ nanosheets sample obtained at reaction temperature of 160 °C denoted as S_160_. (**B**) Sensor responses of α-Fe_2_O_3_ nanosheets to various H_2_S concentrations at 135 °C. (**C**) Sensor responses of the synthesized sensor to 50 ppm of various gases at 135 °C. Figures adapted with permission from [163], Copyright 2018 Elsevier. (**D**) TEM image of the fabricated mesoporous Co_3_O_4_ nanochains. (**E**) Sensor response–recovery curve at 100 ppm of H_2_S gas at 300 °C. (**F**) Sensing response of the prepared Co_3_O_4_ sensor to various gases at 300 °C. Figures adapted with permission from [169], Copyright 2018 Elsevier.

**Figure 14 sensors-19-00905-f014:**
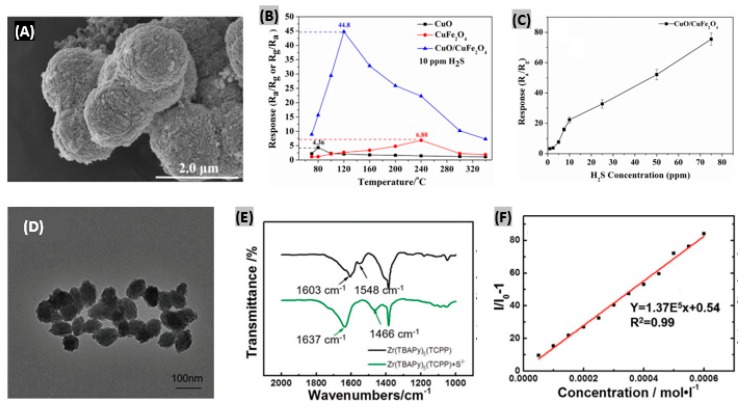
(**A**) FE-SEM image of synthesized CuO/CuFe_2_O_4_ heterostructure. (**B**) Variation of gas sensing responses of CuO microspheres, CuFe_2_O_4_ nanoparticles and CuO/CuFe_2_O_4_ heterostructure with working temperatures. (**C**) H_2_S concentrations dependent sensing responses of CuO/CuFe_2_O_4_ sensor at 240 °C. Figures adapted with permission from [171], Copyright 2018 Elsevier. (**D**) TEM image of the synthesized Zr(TBAPy)_5_(TCPP). (**E**) FTIR spectra of Zr(TBAPy)_5_(TCPP) before and after H_2_S exposure. (**F**) Fluorescent enhancement response of the fabricated sensor under increasing S^2−^ concentration. Figures adapted with permission from [174], Copyright 2018 WILEY-VCH Verlag GmbH & Co. KGaA, Weinheim.

**Figure 15 sensors-19-00905-f015:**
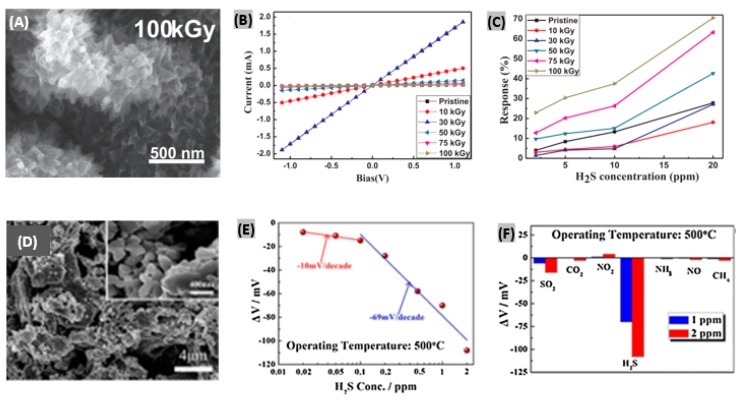
(**A**) FE-SEM image of PANI-Ag film irradiated with EB at 100kGy dose. (**B**) Current-Voltage relationship of PANI-Ag film at pristine and EB irradiated condition. (**C**) Percent sensor responses under different H_2_S concentrations and irradiation doses. Figures adapted with permission from [177], Copyright 2018 Elsevier. (**D**) SEM image of LNO-1 sensing electrode (La_2_NiO_4_). (**E**) Electrode potential difference as a function of the logarithm of H_2_S concentrations (0.02–2 ppm) at 500 °C. (**F**) Selectivity evaluation of the fabricated sensor at 1 and 2 ppm of every test gas. Figures adapted with permission from [180], Copyright 2017 Elsevier.

**Figure 16 sensors-19-00905-f016:**
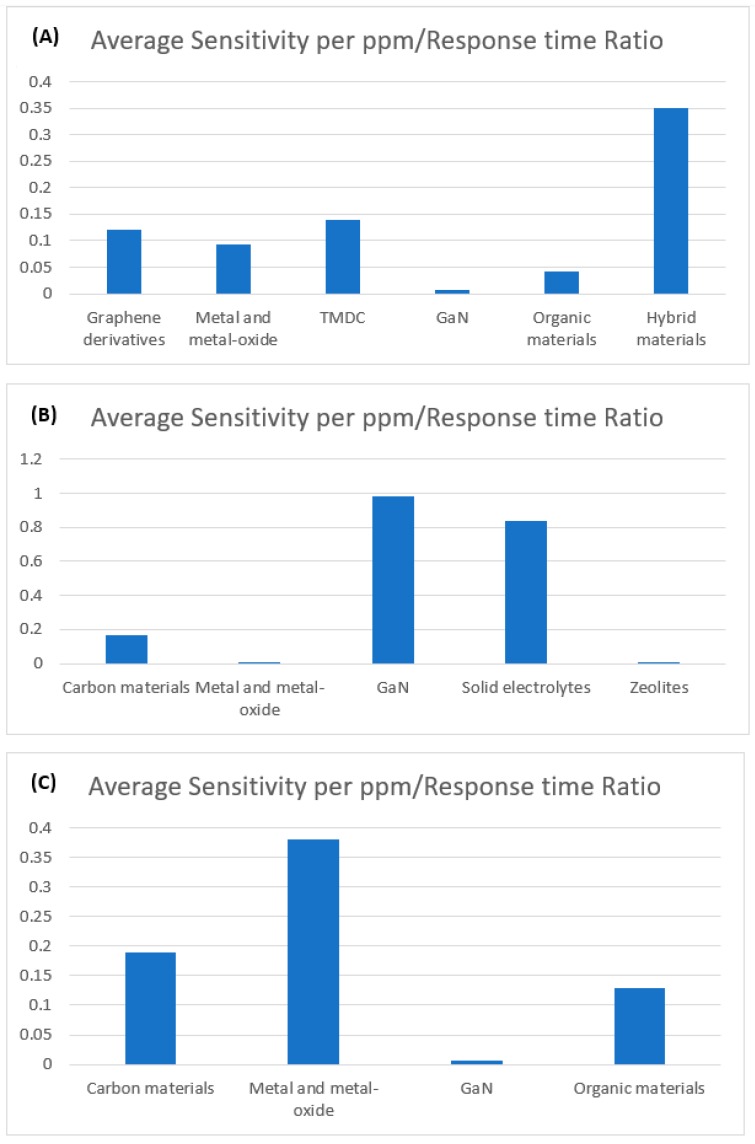
Average sensitivity per ppm/response time ratio comparison among various sensing materials reported in recent years for (**A**) NO_2_ (**B**) SO_2_ and (**C**) H_2_S gas sensors.

**Table 1 sensors-19-00905-t001:** Gas sensing properties of recently developed NO_2_ gas sensors.

Materials	Structure	Operating Temperature (°C)	Concentration (ppm)	Sensitivity/Response	Response Time (s)	Recovery Time (s)	$\frac{Sensitivity\;per\;ppm}{Response\;time}$
SnO_2_/NRGO [104]	Nanosheets	RT	5	1.38	45	168	0.006
Graphene-SnO_2_ [105]	Nanocomposites	150	1	24.7	175	148	0.14
SnO_2_/graphene [106]	Nanocomposites	150 °C	5	26,342	13	Long	405
		RT		171	7 min		0.081
RGO-polythiophene [107]	Thin film	RT	10	26.36	<180	<200	0.015
Ion-Beam Irradiated SnO_2_ [108]	Nanowire	150	2	14.2	292	228	0.025
MoS_2_ [109]	Flakes	RT (UV)	100	27.92	29	350	0.01
	Flakes	100	100	21.56	71	310	0.003
Hierarchical ZnO-RGO [110]	Nanosheets	100	0.05	12	306	450	0.78
MoS_2_/Graphene [111]	Aerogel	200 (microheater)	0.5	9.1	21.6	29.4	0.84
SnO_2_-Polyaniline [112]	Heterostructure thin film	25	50 ppb	5%	5 min	15 min	0.33
RGO/poly(3,4-ethylenedioxythiophene) [113]	Nanocomposite	RT	1	0.05	<180	<70	0.0003
Pd-SnO_2_-RGO [68]	Nanosheets	RT	1	3.92	13	105	0.30
RGO/Au [71]	Nanocomposite	50	5	1.33	132	386	0.002
Mixed p-Type MoS_2_ [76]	Flakes	RT+UV	10	21.78	6.09	146.49	0.36
MoS_2_ [77]	Nanowire networks	60	5	18.1	16	172	0.23
Nb doped-MoSe_2_ [78]	2D Layered	150	3	8.03	<30	-	0.09
PS/WO_3_–Pd60 [84]	Nanorods	RT	2	5.2	10	339	0.26
Polycrystalline PdO [85]	Ultrathin films	175	10	1.63	<500	600–700	0.0003
ZnO [86]	Nanorods	200	100	622	35	206	0.177
ZnO [86]	Bunch of nanowires	200	100	101	17	290	0.06
Microwave-Synthesized NiO [87]	Film	25	3	4991	30	45	55.4
On-chip grown Zn_2_SnO_4_ [89]	Nanowires	200	10	35	<100	<150	0.035
BGaN/GaN superlattice [94]	Double Schottky junction	250	450	31	5	80	0.013
Pt-AlGaN/GaN [95]	HEMT	300	900	33	27 min	-	2.2 × 10^–5^
		600	900	38.5	1.2 min	-	0.0006
SnO_2_-AlGaN/GaN [96]	Heterostructure	250	500 ppb	13%	165	280	0.16
PCDTBT [98]	OFET	RT	10	160	6.5 min	33 min	0.041
Copper Phthalocyanine (CuPc) [100]	Thin film transistor	RT	20	>550	-	>3 days	negligible
MoS_2_-RGO [102]	Nanosheets	160	3	1.23	8	20	0.05
ZnO/poly(3-hexylthiophene) [103]	Nanosheet-nanorod	RT	4	59	<15 min	<45 min	0.02

**Table 2 sensors-19-00905-t002:** Gas sensing properties of recently developed SO_2_ gas sensors.

Materials	Structure	Operating Temperature (°C)	Concentration (ppm)	Sensitivity/Response	Response Time (s)	Recovery Time (s)	$\frac{Sensitivity\;per\;ppm}{Response\;time}$
Polyaniline [144]	Nanoneedles	RT	10	4.2	180	<180	0.0023
Polyaniline-WO_3_ [145]	Nanocomposite	RT	10	10.6	180	180	0.006
SnO_2_ [146]	Thin films	RT	1	138	-	-	-
Au/ZnO [147]	Thin films	RT	10	1.1	20 min	50 min	0.0001
Li_3_PO_4_-Li_2_SO_4_/V_2_O_5_ [148]	Electrolyte film	500	10	30	5 min	10 min	0.01
SnO_2_-TiO_2_ [149]	Composite (75 mol% of TiO_2_)	450	10	55	5 min	>5 min	0.02
g-C_3_N_4_/rGO [150]	2D stacking hybrid	RT	20	0.01 ppm^−1^	204	276	0.5 × 10^−4^
g- C_3_N_4_/rGO [150]	2D stacking hybrid	RT + UV	2	0.0032 ppm^−1^	140	130	2.3 × 10^−5^
Polyaniline [151]	Porous nanofibers	RT	5	4.5%	185	<200	0.005
Au-PANI [112]	Heterostructured thin film	RT	2	300	-	-	-
TiO_2_/rGO [116]	Nanocomposite	RT	1	10.08	73	128	0.14
Ru/Al_2_O_3_/ZnO [119]	Nanosheets	350	25	20	60	6 min	0.013
SrMoO_4_ [120]	nanoflowers	600	2000	−17.2	15.6 min	<30 min	1 × 10^−5^
NiO/SnO_2_ [121]	Thin film	180	500	56	80	70	0.0015
RGO-ZnO on 2DEG AlGaN/GaN [125]	Nanorods	RT	120 ppb	14	120	320	0.98
NASICON-La_0.5_Sm_0.5_FeO_3_ [130]	Thick film electrolyte	275	1	86.5	44	100	1.96
Zirconia-MnNb_2_O_6_ [131]	Electrolyte-electrode	700	5	27	10	>10	0.54
NH_4_^+^ZSM-5 (23) [134]	Zeolites and molecular sieves	RT	4200	6.41	63 min	3 min	0.4 × 10^−6^
CoZn-NCNTs [136]	Nanotube	RT	0.5	8.45%	32	900	0.53
PEDOT-PSS/Y zeolite [137]	Polymer/zeolite composite	27	1000	5	>9.4 min	Longer	0.8 × 10^-5^
MOF-5-NH_2_ [141]	Luminescent probe	RT	0.168	1000 (*luminescence intensity, au*)	<15	-	396

**Table 3 sensors-19-00905-t003:** Gas sensing properties of recently developed H_2_S gas sensors.

Materials	Structure	Operating Temperature (°C)	Concentration (ppm)	Sensitivity/Response	Response Time (s)	Recovery Time (s)	$\frac{Sensitivity\;per\;ppm}{Response\;time}$
In_2_O_3_ [184]	Whiskers	RT	10	35	240	7200	0.015
hc-NiO/N-rGO [185]	Composite	92	100	54.06	100	12	0.0054
rGO/hexagonal WO_3_ [186]	Nanosheets composite	330	10	45	7	55	0.64
Au/Fe_2_O_3_ [187]	Thin films	250	10	6.38	1.65 min	27 min	0.007
α-Fe_2_O_3_ [188]	Micro ellipsoids	350	100	11.7	78	15	0.0015
TiO_2_/ α-Fe_2_O_3_ [189]	Nanorods	300	200	7.4	160	180	2.3 × 10^−4^
Pd/PdO_x_ [190]	Core–shell nanodiscs	200	3	54.9	15	100	1.22
ZnFe_2_O_4_ [191]	Nanosheets	85	5	123	39	34	0.63
SnO_2_/ZnO [192]	Net-like hetero nanostructures	100	5	100	513	98	0.04
SnO_2_-CuO [193]	Coral-like nanocomposite	100	100	38	120	long	0.003
CuO-NiO [194]	Core-shell microspheres	260	100	47	18	29	0.026
Cr-doped WO_3_ [195]	Microsphere	80	0.1 vol. %	89.3	336	300	2.65
SnO_2_ [196]	Multi-tube arrays	RT	5	1.45	14	30	0.02
NiO [197]	Porous nanowall arrays	90	0.001	1.23	49	123	25.1
Cu NPs decorated SWCNTs [198]	Nanotube	RT	5	11%	10	15	0.22
CuO [199]	Porous nanosheets	RT	0.01	1.25	234	76	0.534
PPy-WO_3_ [200]	Nanocomposite films	RT	1	81	360	12,600	0.225
SnO_2_-rGO [201]	Nanofibers	200	5	34	120	550	0.06
SnO_2_-rGO [153]	Nanocomposite	125	100	33.02	209	900	0.002
ZnO-C [154]	Composite nanofibers	250	50	102	-	-	-
Pt-gated AlGaN/GaN [155]	HEMT	250	90	112	219	507	0.006
α-Fe_2_O_3_ [164]	Nanosheets	135	5	5.8	10	45	0.116
ZnO/CuO [165]	Nanotube	50	5	25	37	94	0.135
In_2_O_3_ [166]	Porous nanoparticles	25	1	26268.5	>200	>200	131.3
Cu_2_O-doped SnO_2_ [167]	Nanorod	RT	5	30	21	204	0.29
p-type Co_3_O_4_ [169]	Mesoporous nanochains	300	100	4.5	46	24	0.001
SnO_2_-CuWO_4_ [170]	Mesoporous layers	100	20	2 × 10^6^ (sensor signal)	2.5 min	7.3 min	-
Chitosan-IL [178]	Film	80	100	200%	>15	-	0.13
Cu^2+^-Doped SnO_2_/Poly pyrrole [179]	Hybrid nanospheres	RT	0.3	9	7	14	4.28

**Table 4 sensors-19-00905-t004:** The adsorption energy (eV), Shortest adsorption distance (Å) and charge transfer (e) between NO_2,_ SO_2,_ H_2_S and various sensing materials at the most stable adsorption configuration based on DFT.

Materials	Target Gas	Adsorption Energy (eV)	Shortest Adsorption Distance (Å)	Charge Transfer (e)
Pristine Arsenene [202]	NO_2_	−0.4378	2.955	−0.187
B-doped Arsenene [202]	NO_2_	−1.913	1.562	−0.273
N-doped Arsenene [202]	NO_2_	−0.4502	2.506	−0.163
Pristine Arsenene [202]	SO_2_	−0.3413	2.957	−0.192
B-doped Arsenene [202]	SO_2_	−1.0733	1.961	−0.141
N-doped Arsenene [202]	SO_2_	−0.8597	2.278	−0.251
GeSe monolayer (Ge_Top_) [203]	SO_2_	−0.58	2.86	−0.2788
GeSe monolayer (Ge_Top_) [203]	NO_2_	−2.24	2.29	−0.464
GeSe monolayer (Se_Top_) [203]	SO_2_	−0.52	2.84	−0.2321
GeSe monolayer (Se_Top_) [203]	NO_2_	−1.97	3.09	−0.2505
Borophene (buckled) [204]	NO_2_	1.75	1.56	0.76
Borophene (line-defective) [204]	NO_2_	1.80	1.57	0.89
Monolayer C_3_N [205]	NO_2_	−0.79	2.89	−0.388
Monolayer C_3_N [205]	H_2_S	−0.23	3.39	−0.004
Monolayer C_3_N [205]	SO_2_	−0.62	2.84	−0.247
Blue Phosphorene [206]	H_2_S	−0.242	3.2	0.037
Blue Phosphorene [206]	SO_2_	−0.247	3.0	−0.138
C_2_H_4_Li [207]	NO_2_	4.07	1.90	−0.77
C_2_H_4_Li [207]	SO_2_	3.09	1.90	−0.38
Ni-MoS_2_ monolayer [208]	H_2_S	−1.319	2.205	0.254
Ni-MoS_2_ monolayer [208]	SO_2_	−1.382	2.059	−0.016
2D Tetragonal GaN [211]	NO_2_	−0.673	2.066	−0.108
Graphitic GaN sheet [212]	NO_2_	−0.493	2.44	−0.081
Graphitic GaN sheet [212]	SO_2_	−1.06	1.79	−0.209
Graphitic GaN sheet [212]	H_2_S	−0.446	2.89	0.139

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
