# Peer review of "Recent Advances in Electrochemical Sensors for Detecting Toxic Gases: NO2, SO2 and H2S"

_sensors, 2019, doi:10.3390/s19040905_

Round 1
Reviewer 1 Report
The manuscript is a review of recent advances in electrochemical sensors for detection of toxic gases (i.e. NO2, SO2 and H2S). The topic is relevant and worth of revision and therefore the paper might be an important contribution to the field. However, there are still missing some crucial information that could improve the paper quality. In my opinion, the paper may be accepted after major revisions. Below, there are few specific comments:
- In the introduction section, the authors should provide more information about toxic concentration range of the selected gases relating to different scenarios. For instance: hydrogen sulfide can be lethal within seconds in the concentration of 1000 ppm and it is particularly dangerous in petroleum extraction platforms.
- In introduction section, the authors could at least cite other detection techniques, such as optical (e.g. fluorescence, absorbance) or separation methods (CG, HPLC) and discuss the advantages and disadvantages of electrochemical detection.
- Although the authors described the detection of each gas by each electrochemical material, I believe it is missing more discussion on applicability of such sensors in real-world scenarios. If there is not field validation, the authors could discuss the reasons within the paper.
- Line 83: what is the response parameter?
- Table 1 (and others): Column 4 is concentration or limit of detection?
- Some sensors described work at high temperature. Is it a disadvantage? The authors could include more discussion on this matter.
- In many reviews found in the literature, is not usual any discussion about the calibration of gas sensors using standard gas mixtures. I consider this an important topic, specially related to gaseous solutions. Maybe the authors could consider to insert some information on this matter.
- In the last two lines of the introduction section, it is claimed that the goal is shine a light on the development of toxic gas detection and that is a vital technology for the emerging internet of things era. However, this term is no longer mentioned and there is no further discussion on this. I did not really understand what the authors meant.
- Could materials doped with Pd and employed for NO2 detection suffer from H2S or SO2 interference, as palladium-sulfur interaction is very strong.
Author Response
Thank you very much for reviewing our manuscript. We greatly appreciate you for the complimentary comments and suggestions. The followings are our point-by-point responses:
Comment: “In the introduction section, the authors should provide more information about toxic concentration range of the selected gases relating to different scenarios. For instance: hydrogen sulfide can be lethal within seconds in the concentration of 1000 ppm and it is particularly dangerous in petroleum extraction platforms.”
Response: We have included more information about toxic concentration range of the mentioned gases in the introduction section. (Page 2, Lines 68,79; Page 3, Line 89).
Comment: “In introduction section, the authors could at least cite other detection techniques, such as optical (e.g. fluorescence, absorbance) or separation methods (CG, HPLC) and discuss the advantages and disadvantages of electrochemical detection.”
Response: We have cited optical, acoustic and gas chromatographic detection techniques. Then, we have discussed the advantages and disadvantages of electrochemical detection. (Page 3, Lines 96-103).
Comment: “Although the authors described the detection of each gas by each electrochemical material, I believe it is missing more discussion on applicability of such sensors in real-world scenarios. If there is not field validation, the authors could discuss the reasons within the paper.”
Response: We have added a new paragraph (Page 27, Line 801: 7. “Toxic gas sensors in Internet of Things (IoT) applications”). The real-world applications of the gas sensors are discussed in there and some related references have been cited.
Comment: “Line 83: what is the response parameter?”
Response: The response parameter used there is the ratio between Resistance in air and Resistance in target gas (Ra/Rg). Since Rg decreases with increasing NO2 concentration, (Ra/Rg) ratio increases with concentration.
Comment: “Table 1 (and others): Column 4 is concentration or limit of detection?”
Response: In Table 1, 2 and 3, column 4 is the concentration at which sensitivity of column 5 is obtained. Generally, those are the most stable responses for the mentioned fabricated gas sensors.
Comment: “Some sensors described work at high temperature. Is it a disadvantage? The authors could include more discussion on this matter.”
Response: Sensors having high working temperature are generally employed in industrial, mining and space applications. We have added some discussion on it in the introduction section. (Page 3, Lines 100-103).
Comment: “In many reviews found in the literature, is not usual any discussion about the calibration of gas sensors using standard gas mixtures. I consider this an important topic, specially related to gaseous solutions. Maybe the authors could consider to insert some information on this matter.”
Response: We have added a new paragraph to discuss about the calibration of gas sensors (Page 27, Line 790: 6. “Calibration of the toxic gas sensors”). Some related references have been cited as well.
Comment: “In the last two lines of the introduction section, it is claimed that the goal is shine a light on the development of toxic gas detection and that is a vital technology for the emerging internet of things era. However, this term is no longer mentioned and there is no further discussion on this. I did not really understand what the authors meant.”
Response: We have added a new paragraph to discuss more on the internet of things (IoT) applications of toxic gas detection. (Page 27, Line 801: 7. “Toxic gas sensors in Internet of Things (IoT) applications”).
Comment: “Could materials doped with Pd and employed for NO2 detection suffer from H2S or SO2 interference, as palladium-sulfur interaction is very strong.”
Response: In our review, there are 2 instances where Pd had been used to detect NO2. In reference [84], porous silicon/WO3 nanorods based sensor functionalized with Pd NPs is selective to NO2 against SO2 according to their selectivity test. However, H2S and other sulfur containing gases were not present in the test gas list. And, in case of Pd-SnO2-RGO hybrid (reference [67]), the authors didn’t include any S- containing gases in their selectivity test. So, we have added some discussion (Page 3, Line 135-137) and cited a reference paper on palladium-sulfur interaction (Reference 68).
Reviewer 2 Report
Dear Sir,
The manuscript presents an extensive review on the subject for electrochemical detection methods for the three mentioned harmful gases. Although the list of references is quite exhaustive, there are still some remarks that should have been backed by illustrative references. For example, the statement “The major sources of NO2 are from combustion of fuels such as certain coals and oil [1,2], biomass burning due to extreme heat of lightning during thunderstorms, and nitrogen fixation by microorganisms due to agricultural fertilization.” has references for only the first source of nitrogen dioxide generation. Biomass burning or nitrogen fixation processes should have also been illustrated by proper references. Same observation for Sox or H2S and other similar situations throughout the text.
I think that the last sentence of the Introduction paragraph (“The goal is to shine a light on the future development trend of toxic gas detection, a vital technology for the emerging Internet of Things era”) since a direct connection between toxic gas detection devices and the concept of “Internet of Things” is not obvious. But if the authors wish to maintain this statement, then they will have to demonstrate the existence and importance of such a link (maybe in paragraph nr. 6 “Future perspectives and conclusion”).
Since the manuscript is mostly a review of recent literature on the subject, after the abstract a content listing the successive paragraphs would have been welcomed, so the reader could apprehend rapidly the bibliographic material systematization.
Since the manuscript wish to be a review of the most recent advances in the field of modern sensors for the detection of NO2, SO2and H2S, than important references should not have been missed (e.g. Detection of ultra-low concentration NO2 in complex environment using epitaxial graphene sensors, Christos Melios, Vishal Panchal, Kieran Edmonds, Arseniy Lartsev, Rositsa Yakimova, and Olga Kazakova, ACS Sensors, 2018, 3(9), 1666-1674; Sulfuration-desulfuration reaction sensing-effect of intrinsic ZnO nanowires for high-performance H2S detection, Haiyun Huang, Pengcheng Xu, Dan Zheng, Chuanzhao Chen and Xinxin Li, J. Mater. Chem. A, 2015, 3, 6330-6339; S. Das, S. Rana, S.M. Mursalin, P. Rana, A. Sen, Sonochemically prepared nanosized BiFeO3 as novel SO2 sensor, Sensors and Actuators B: Chemical, 2015, 218, 122-127 etc.).
Since the manuscript is a review and there were previously such reviews made in this field, perhaps that such previous reviews should have been mentioned in the Introduction paragraph (e.g. Swain S.K., Barik S., Das R. (2018) Nanomaterials as Sensor for Hazardous Gas Detection. In: Martínez L., Kharissova O., Kharisov B. (eds) Handbook of Ecomaterials. Springer, Cham; Gas Sensors: A Review, Zainab Yunusa, Mohd. Nizar Hamidon, Ahsanul Kaiser, Zaiki Awang, Sensors & Transducers, Vol. 168, Issue 4, April 2014, pp. 61-75; M.W.G. Hoffmann, O. Casals, A.E. Gad, L. Mayrhofer, C. Fàbrega L. Caccamo, F. Hernández-Ramírez, G. Lilienkamp, W. Daum, M. Moseler, H. Shen, A. Waag, J.D. Prades, Novel Approaches towards Highly Selective Self-Powered Gas Sensors, Procedia Engineering, 2015, 120, 623-627; Shashikant V. Lahade, Pravin D. Pardhi, Gas Sensing Technologies: Review, Scope and Challenges, International Journal of Recent Trends in Engineering & Research (IJRTER), 2018, 4(2), 108-115; Swati Sharma, Marc Madou, Review: A new approach to gas sensing with nanotechnology, Phil. Trans. R. Soc. A, 2012, 370, 2448–2473; M.M. Rana, D.S. Ibrahim, M.R.M. Asyraf, S. Jarin, A. Tomal, A review on recent advances of CNTs as gas sensors, Sensor Review, 2017, 37(2), 127-136 etc.).
There should be some kind of order in Table 1: group the sensing method either by working temperature, decreasing sensitivity, response time or type of structure (same observation for Tables 2 and 3). Table 4 should be ordered in function of the target gas.
Minor comments:
Keep the same font size throughout the text (e.g. “keratoconjunctivitis” is written with a larger font).
Eliminate typos (e.g. “such as- carbon …” – no need for the dash sign).
Keep the same color for bibliographic references throughout the text (some are blue, some are black).
Put “vs” abbreviation for “versus” in italic.
Don’t leave a blank space between value and Celsius degree (e.g. “125°C”, not “125 °C”).
Be consistent in the references list with all the citations (it’s either Initial + Name or complete full names for all listed papers, not both).
“Acknowledgement” paragraph is number 7.

Author Response
Thank you very much for reviewing our manuscript. We greatly appreciate you for the complimentary comments and suggestions. The followings are our point-by-point responses:
Major comments:
Comment: “The manuscript presents an extensive review on the subject for electrochemical detection
methods for the three mentioned harmful gases. Although the list of references is quite
exhaustive, there are still some remarks that should have been backed by illustrative
references. For example, the statement “The major sources of NO2 are from combustion of
fuels such as certain coals and oil [1,2], biomass burning due to extreme heat of lightning
during thunderstorms, and nitrogen fixation by microorganisms due to agricultural
fertilization.” has references for only the first source of nitrogen dioxide generation.
Biomass burning or nitrogen fixation processes should have also been illustrated by proper
references. Same observation for Sox or H2S and other similar situations throughout the
text.”
Response: We have added references for biomass burning and nitrogen fixation processes (Page 2, Line 71,72). In case of SO2 or H2S, references have been given altogether for sources or effects.
Comment: “I think that the last sentence of the Introduction paragraph (“The goal is to shine a light on
the future development trend of toxic gas detection, a vital technology for the emerging
Internet of Things era”) since a direct connection between toxic gas detection devices and
the concept of “Internet of Things” is not obvious. But if the authors wish to maintain this
statement, then they will have to demonstrate the existence and importance of such a link
(maybe in paragraph nr. 6 “Future perspectives and conclusion”).”
Response: We have added a new paragraph to discuss more on the internet of things (IoT) applications of toxic gas detection. (Page 27, Line 801: 7. “Toxic gas sensors in Internet of Things (IoT) applications”). Various applications of the toxic gas sensors in IoT platforms are mentioned in there and some related references have been cited.
Comment: “Since the manuscript is mostly a review of recent literature on the subject, after the abstract a content listing the successive paragraphs would have been welcomed, so the reader could
apprehend rapidly the bibliographic material systematization.”
Response: There was an outline listing successive paragraphs in our submitted manuscript. We think it was removed by the editor during layout editing. So, we have re-added that list of contents (Page 1, Line 30) listing the successive paragraphs.
Comment: “Since the manuscript wish to be a review of the most recent advances in the field of modern
sensors for the detection of NO2, SO2and H2S, than important references should not have
been missed (e.g. Detection of ultra-low concentration NO2 in complex environment using
epitaxial graphene sensors, Christos Melios, Vishal Panchal, Kieran Edmonds, Arseniy Lartsev, Rositsa Yakimova, and Olga Kazakova, ACS Sensors, 2018, 3(9), 1666-1674;
Sulfuration-desulfuration reaction sensing-effect of intrinsic ZnO nanowires for highperformance H2S detection, Haiyun Huang, Pengcheng Xu, Dan Zheng, Chuanzhao Chen and Xinxin Li, J. Mater. Chem. A, 2015, 3, 6330-6339;
S. Das, S. Rana, S.M. Mursalin, P. Rana, A. Sen, Sonochemically prepared nanosized BiFeO3 as novel SO2 sensor, Sensors and Actuators B: Chemical, 2015, 218, 122-127 etc.).”
Response: We have included all the mentioned papers as reference no. [66], [166] and [123] respectively in our manuscript.
Comment: “Since the manuscript is a review and there were previously such reviews made in this field,
perhaps that such previous reviews should have been mentioned in the Introduction paragraph (e.g. Swain S.K., Barik S., Das R. (2018) Nanomaterials as Sensor for Hazardous Gas Detection. In: Martínez L., Kharissova O., Kharisov B. (eds) Handbook of Ecomaterials. Springer, Cham;
Gas Sensors: A Review, Zainab Yunusa, Mohd. Nizar Hamidon, Ahsanul Kaiser, Zaiki Awang, Sensors & Transducers, Vol. 168, Issue 4, April 2014, pp. 61-75;
M.W.G. Hoffmann, O. Casals, A.E. Gad, L. Mayrhofer, C. Fàbrega L. Caccamo, F. Hernández-Ramírez, G. Lilienkamp, W. Daum, M. Moseler, H. Shen, A. Waag, J.D. Prades, Novel Approaches towards Highly Selective Self-Powered Gas Sensors, Procedia Engineering, 2015, 120, 623-627;
Shashikant V. Lahade, Pravin D. Pardhi, Gas Sensing Technologies: Review, Scope and Challenges, International Journal of Recent Trends in Engineering & Research (IJRTER), 2018, 4(2), 108-115;
Swati Sharma, Marc Madou, Review: A new approach to gas sensing with nanotechnology, Phil. Trans. R.
Soc. A, 2012, 370, 2448–2473;
M.M. Rana, D.S. Ibrahim, M.R.M. Asyraf, S. Jarin, A. Tomal, A review on recent advances of CNTs as gas sensors, Sensor Review, 2017, 37(2), 127-136 etc.).”
Response: We have included all the mentioned review papers as reference no. [60], [19], [61], [62], [63] and [25] respectively in our manuscript.
Comment: “There should be some kind of order in Table 1: group the sensing method either by working
temperature, decreasing sensitivity, response time or type of structure (same observation for
Tables 2 and 3). Table 4 should be ordered in function of the target gas.”
Response: Since we have reviewed each toxic gas detection by categorizing into sensing materials, the tables have been arranged according to sensing materials. For instance, graphene and its derivative based sensors are grouped together, then metal-oxide based sensors are listed together and so on.
Minor comments:
Comment: “Keep the same font size throughout the text (e.g. “keratoconjunctivitis” is written with a
larger font).
Eliminate typos (e.g. “such as- carbon …” – no need for the dash sign).
Keep the same color for bibliographic references throughout the text (some are blue, some are black).
Put “vs” abbreviation for “versus” in italic.
Don’t leave a blank space between value and Celsius degree (e.g. “125°C”, not “125 °C”).
Be consistent in the references list with all the citations (it’s either Initial + Name or complete full names for all listed papers, not both).
“Acknowledgement” paragraph is number 7”
Response: We have adjusted all the mentioned suggestions in our manuscript. Thank you very much for these detailed suggestions.
Round 2
Reviewer 1 Report
The authors have provided all responses and modification within the text. In my opinion, the paper is suitable for publishing.